# Response of dust emissions in southwestern North America to 21st century trends in climate, CO$_2$ fertilization, and land use: Implications for air quality

Yang Li[1], Loretta J. Mickley[1], Jed O. Kaplan[2]

[1]John A. Paulson School of Engineering and Applied Sciences, Harvard University, Cambridge, MA, USA

[2]Department of Earth Sciences, The University of Hong Kong, Hong Kong, China

*Correspondence to*: Yang Li (yangli@seas.harvard.edu)

**Abstract.** Climate models predict a shift toward warmer and drier environments in southwestern North America. The consequences of such a shift for dust mobilization and dust concentration are unknown, but could have large implications for human health, given connections between dust inhalation and disease. Here we link a dynamic vegetation model (LPJ-LMfire) to a chemical transport model (GEOS-Chem) to assess the impacts of future changes in three factors – climate, CO$_2$ fertilization, and land use practices – on vegetation in this region. From there we investigate the impacts of changing vegetation on dust mobilization and assess the net effect on fine dust concentration (defined as dust particles less than 2.5 microns in diameter) on surface air quality. We find that surface temperatures in southwestern North America warm by 3.3 K and precipitation decreases by nearly 40% by 2100 in the most extreme warming scenario (RCP8.5) in spring (March, April, and May), the season of greatest dust emissions. Such conditions reveal an increased vulnerability to drought and vegetation die-off. Enhanced CO$_2$ fertilization, however, offsets the modeled effects of warming temperatures and rainfall deficit on vegetation in some

areas of the southwestern United States. Considering all three factors in RCP8.5 scenario, dust
concentrations decrease over Arizona and New Mexico in spring by the late 21$^{st}$ century due to
greater $CO_2$ fertilization and a more densely vegetated environment, which inhibits dust
mobilization. Along Mexico's northern border, dust concentrations increase as a result of the
intensification of anthropogenic land use. In contrast, when $CO_2$ fertilization is not considered in
the RCP8.5 scenario, vegetation cover declines significantly across most of the domain by 2100,
leading to widespread increases in fine dust concentrations, especially in southeastern New Mexico
(up to ~2.0 $\mu$g m$^{-3}$ relative to the present day) and along the border between New Mexico and
Mexico (up to ~2.5 $\mu$g m$^{-3}$). Our results have implications for human health, especially for the
health of the indigenous people who make up a large percentage of the population in this region.

## 1 Introduction

The arid and semi-arid region covering southwestern United States and northwestern Mexico is characterized by large concentrations of soil-derived dust particles in the lower atmosphere, especially in spring (Hand et al., 2016). By causing respiratory and cardiovascular diseases, fine dust particles – i.e., those particles with diameter less than 2.5 microns – can have negative effects on human health (Tong et al., 2017; Meng and Lu, 2007; Gorris et al., 2018). A key question is to what extent climate change and other factors will influence future dust concentrations in this region, which we define here as southwestern North America. In this study, we use a suite of models to predict the future influence of three factors – climate change, increasing $CO_2$ fertilization, and land use change – on vegetation in this region, and assess the consequences for dust mobilization and dust concentrations.

Wind speed and vegetation cover are two key factors that determine soil erodibility and dust emissions. Wind gusts mobilize dust particles from the Earth's surface, while vegetation constrains dust emissions by reducing the extent of bare land and preserving soil moisture (Zender et al., 2003). The high temperatures and reduced soil moisture characteristic of drought play an important role in dust mobilization, since loss of vegetative cover during drought increases soil erosion (Archer and Predick, 2008; Bestelmeyer et al., 2018).

Southwestern North America is covered by desert grassland, perennial grassland, savanna, desert scrub, and grassy shrublands or woodlands (McClaran and Van Devender, 1997). In recent decades, a gradual transition from grasslands to shrubland has been observed across much of this region, with increased aridity, atmospheric $CO_2$ enrichment, and livestock grazing all possibly playing a role in this trend (Bestelmeyer et al., 2018). Future climate change may further prolong this transition, especially since shrubs fare better than grasses under a climate regime characterized

by large fluctuations in annual precipitation (Bestelmeyer et al., 2018; Edwards et al., 2019).
Climate models predict a warmer and drier environment in southwestern North America through
the 21st century, with more frequent and severe drought (Seager and Vecchi, 2010; MacDonald,
2010; Stahle, 2020; Prein et al., 2016; Williams et al., 2020). Such conditions would decrease
vegetative cover and allow for greater dust mobilization. On the other hand, elevated $CO_2$
concentrations in the future atmosphere could increase photosynthesis and decrease transpiration
of some vegetation species, allowing for more efficient water use and enhancing growth (Poorter
and Perez-Soba, 2002; Polley et al., 2013). Anthropogenic land use practices – e.g., agriculture,
human settlement, and urban sprawl – have changed dramatically over the southwestern North
America in recent decades, with Arizona and New Mexico showing decreasing cropland area and
northern Mexico experiencing increasing pasture area (Figure S1). Future land use practices could
also influence the propensity for dust mobilization by disturbing crustal biomass (e.g., Belnap and
Gillette, 1998).
Previous studies have investigated the relative importance of climate, $CO_2$ fertilization,
and/or land use in present-day and future dust emissions and concentrations, sometimes with
contradictory results. For example, Woodward et al., 2005 predicted a tripling of the global dust
burden by 2100 relative to the present day, while other studies suggested a decrease in the global
dust burden (e.g., Harrison et al., 2001, Mahowald and Luo, 2003 and Mahowald et al., 2006).
These estimates of future dust emissions depended in large part on the choice of model applied, as
demonstrated by Tegen et al., 2004.
In southwestern North America, a few recent studies examined statistical relationships
between observed present-day dust concentrations and meteorological conditions or leaf area index
(LAI). Hand et al., 2016 found that fine dust concentrations in spring in this region correlated with
the Pacific Decadal Oscillation (PDO), indicating the importance of large-scale climate patterns in
the mobilization and transport of regional fine dust. Tong et al., 2017 further determined that the
observed 240% increase in the frequency of windblown dust storms from 1990s to 2000s in the
southwestern United States was likely associated with the PDO. Similarly, Achakulwisut et al.,
2017 found that the 2002–2015 increase in average March fine dust concentrations in this region
was driven by a combination of positive  PDO conditions and phase of the El Nino-Southern
Oscillation. More recently, Achakulwisut et al., 2018 identified the Standardized Precipitation-
Evapotranspiration Index as a useful indicator of present-day dust variability. Applying that metric
to an ensemble of future climate projections, these authors predicted increases of 26-46% in fine
dust concentrations over the U.S. Southwest in spring by 2100. In contrast, Pu and Ginoux, 2017
found that the frequency of extreme dust days decreases slightly in spring in this region due to
reduced extent of bare land under 21$^{st}$ century climate change.

These regional studies relied mainly on statistical models that relate local and/or large scale

meteorological conditions to dust emissions in southwestern North America. Pu and Ginoux, 2017
also considered changing LAI in their model, but these dust-LAI relationships were derived from
a relatively sparse dataset, casting some uncertainty on the results (Achakulwisut et al., 2018).  In
this study, we investigate the effects of climate change, increasing $CO_2$ fertilization, and future
land use practices on vegetation in southwestern North America, and we examine the response of
dust mobilization due to these changes in vegetation. With regard to climate, we examine whether
a shift to warmer, drier conditions by 2100 enhances dust mobilization in this region by reducing
vegetation cover and exposing bare land. To that end, we couple the LPJ-LMfire dynamic
vegetation model to the chemical transport model GEOS-Chem to study vegetation dynamics and
dust mobilization under different conditions and climate scenarios, allowing consideration of
several factors driving future dust mobilization in the southwestern North America. We focus on
fine dust particles in springtime (March, April, and May), because it is the season of highest dust
concentrations in the southwestern U.S. (Hand et al., 2017). Given the deleterious impacts of
airborne dust on human health, our dust projections under different climate scenarios have value
for understanding the full array of potential consequences of anthropogenic climate change.

**2  Methods**
We examine dust mobilization in southwestern North America, here defined as 25°N –
37°N, 100°W – 115°W (Figure 1), during the late-21st century under scenarios of future climate
and land use based on two Representative Concentration Pathways (RCPs). RCP4.5 and RCP8.5
capture two possible climate trajectories over the 21st century, beginning in 2006. RCP4.5
represents a scenario of moderate future climate change with gradual reduction in greenhouse gas
(GHG) emissions after 2050 and a radiative forcing at 2100 relative to pre-industrial values of +4.5
W m$^{-2}$, while RCP8.5 represents a more extreme scenario with continued increases in GHGs
throughout the 21st century and a radiative forcing of +8.5 W m$^{-2}$ at 2100. For each RCP, we
investigate the changes in vegetation for three cases: 1) an all-factor case that includes changes in
climate, land use, and $CO_2$ fertilization; 2) a fixed-$CO_2$ case that includes changes in only climate
and land use; and 3) a fixed-land use case that includes changes in only climate and $CO_2$
fertilization.
We use LPJ-LMfire, a dynamic global vegetation model, to estimate changes in vegetation
under future conditions (Pfeiffer et al., 2013). Meteorology to drive LPJ-LMfire is taken from the
Goddard Institute for Space Studies (GISS) climate model (Nazarenko et al., 2015).  Using the
GEOS-Chem emission component (HEMCO), we then calculate dust emissions based on the LPJ-
generated vegetation area index (VAI) for all scenarios. We apply the resulting dust emissions to
the global chemical transport model GEOS-Chem to simulate the distribution of fine dust across
the southwestern North America.

**2.1   GISS Model E**

Present-day and future meteorological fields for RCP4.5 and RCP8.5 are simulated by the

GISS Model E climate model (Nazarenko et al., 2015), configured for Phase 5 of the Coupled
Model Intercomparison Project (CMIP5; https://esgf-node.llnl.gov/search/cmip5/, last accessed on
17 July 2020). The simulations cover the years 1801 to 2100 at a spatial resolution of 2° latitude x
2.5° longitude. Changes in climate in the GISS model are driven by increasing greenhouse gases.
In RCP4.5, $CO_2$ concentrations increase to 550 ppm by 2100; in RCP8.5 the $CO_2$ increases to 1960
ppm ((Meinshausen et al., 2011).

Under RCP4.5, the GISS model predicts a slight increase of 0.45 K in springtime mean

surface temperatures and an increase in mean precipitation by ~17% over the southwestern North
America by the 2100 time slice (2095-2099), relative to the present day (2011-2015). In contrast,
under RCP8.5, the 5-year mean springtime temperature increases significantly by 3.29 K by 2100
and mean precipitation decreases by ~39%. The spatial distributions of the changes in temperature
and precipitation by 2100 under RCP8.5 are presented in the Supplement (Figure S2). In addition,
lightning strike densities decrease by ~0.006 strikes $km^{-2}$ $d^{-1}$ over Arizona in RCP4.5, but increase
by the same magnitude in this region in RCP8.5 (Li et al., 2020). Lightning strikes play a major
role for wildfire ignition in this region, while wildfires may influence landscape succession (e.g.,
Bodner and Robles, 2017). Finally, future surface wind speeds do not change significantly under
RCP4.5, but increase slightly by ~4% across southwestern North America under RCP8.5 by 2100
(not shown). The increasing winds in RCP8.5 will influence the spread of fires in our study, but
will not affect the simulated dust fluxes directly, as described in more detail below. Compared to
those from other climate models, the GISS projections of climate change in southwestern North
America are conservative (Ahlström et al., 2012; Sheffield et al., 2013), implying that our
predictions of the impact of climate change on dust mobilization may also be conservative.
In our study, we do not specifically track drought frequency under future climate, as the
definition of drought is elusive (Andreadis et al., 2005; Van Loon et al., 2016). Nonetheless, the
meteorological conditions predicted in the RCP8.5 scenario for 2100 align with previous studies
projecting increased risk of drought in this region (e.g., Williams et al., 2020), and as we shall see,
such conditions, in the absence of $CO_2$ fertilization, result in decreased vegetation and greater dust
mobilization.
**2.2   LPJ-LMfire**
LPJ-LMfire is a dynamic vegetation model that includes a process-based representation of
fire (Pfeiffer et al., 2013). Input to LPJ-LMfire includes meteorological variables, soil
characteristics, land use, and atmospheric $CO_2$ concentrations, and the model then simulates the
corresponding vegetation structure, biogeochemical cycling, and wildfire at a spatial resolution of
0.5° latitude x 0.5° longitude. Here "vegetation structure" refers to vegetation types and the spatial
patterns in landscapes.
More specifically, LPJ-LMfire simulates the impacts of photosynthesis, evapotranspiration,
and soil water dynamics on vegetation structure and the population densities of different plants
functional types (PFTs). The model considers the coupling of different ecosystem processes, such
as the interactions between $CO_2$ fertilization, evapotranspiration, and temperature as well as the
competition among different PFTs for water resources (e.g., precipitation, surface runoff, and
drainage). The different PFTs in LPJ-LMfire respond differently to changing $CO_2$, with $CO_2$
enrichment preferentially stimulating photosynthesis in woody vegetation and $C_3$ grasses
compared to $C_4$ grasses (Polley et al., 2013). Wildfire in LPJ-LMfire depends on lightning ignition,
and the simulation considers multiday burning, coalescence of fires, and the spread rates of
different vegetation types. The effects of changing fire activity on vegetation cover are then taken
into account (Pfeiffer et al., 2013; Sitch et al., 2003; Chaste et al., 2019). Li et al., 2020 predicted
a ~50% increase in fire-season area burned by 2100 under scenarios of both moderate and intense
future climate change over the western United States. However, the effects of changing fire on
vegetation cover are insignificant in the grass and bare ground-dominated ecosystems of the desert
Southwest, where the low biomass fuels cannot support extensive spread of fires.

For this study we follow Li et al., 2020, in linking meteorology from GISS-E2-R to LPJ-

LMfire in order to capture the effects of climate change on vegetation. Meteorological fields from
the GISS model include monthly mean surface temperature, diurnal temperature range, total
monthly precipitation, number of days in the month with precipitation greater than 0.1 mm,
monthly mean total cloud cover fraction, and monthly mean surface wind speed. Monthly mean
lightning strike density, calculated using the GISS convective mass flux and the empirical
parameterization of Magi, 2015, is also applied to LPJ-LMfire. To downscale the 2° x 2.5° GISS
meteorology to finer resolution for LPJ-LMfire, we calculate the 2010-2100 monthly anomalies
relative to the average over the 1961-1990 period, and then add these anomalies to an
observationally based climatology (Pfeiffer et al., 2013). LPJ-LMfire then simulates the response
of natural vegetation to the 21[st] century trends in these meteorological fields and to increasing $CO_2$.
We apply the same changes in $CO_2$ concentrations as those applied to the GISS model.

We overlay the changes in natural land cover with future land use scenarios from CMIP5

(Hurtt et al., 2011; http://tntcat.iiasa.ac.at/RcpDb/, last accessed on 17 July 2020).  Such land use
includes agriculture, human settlement, and urban sprawl, all of which result in habitat loss and
the fragmentation of forested landscapes. Present-day land use prepared for CMIP5 is taken from
the HYDE database v3.1 (Goldewijk, 2001; Goldewijk et al., 2010), which in turn is based on
array of sources, including satellite observations and government statistics. In our simulations, fire
is not allowed to occur on cropland and rangeland, so we do consider some land management. On
the other hand, our model does not account for the density of livestock on rangeland, which when
mismanaged, can lead to reduction of vegetation cover and enhanced dust emissions. In RCP8.5,
the extent of cropland and pasture cover increases by ~30% in Mexico but decreases by 10-20%
over areas along Mexico's northern border in the U.S. (Hurtt et al., 2011). Only minor changes in
land use practices by 2100 are predicted under RCP4.5 (Hurtt et al., 2011).

We perform global simulations with LPJ-LMfire on a 0.5° x 0.5° grid for the two RCPs

from 2006-2100, and analyze results over southwestern North America, where dust emissions are
especially high.  For each RCP we consider the effects of changing climate on land cover, as well
as the influence of anthropogenic land use change and $CO_2$ fertilization.  The LPJ-LMfire
simulations yield monthly timeseries of the leaf area indices (LAI) and fractional vegetation cover
($\sigma_v$) for nine plant functional types (PFTs): tropical broadleaf evergreen, tropical broadleaf
raingreen, temperate needleleaf evergreen, temperate broadleaf evergreen, temperate broadleaf
summergreen, boreal needleleaf evergreen, and boreal summergreen trees, as well as $C_3$ and $C_4$
grasses.  We further discuss the LPJ-LMfire present-day land cover in the Supplement.
**2.3   VAI calculation**

Vegetation constrains dust emissions in two ways: 1) by competing with bare ground as a

sink for atmospheric momentum, which results in less drag on erodible soil (Nicholson et al., 1998;
Raupach, 1994); and 2) by enhancing soil moisture through plant shade and root systems (Hillel,
1982). Here we implement the dust entrainment and deposition (DEAD) scheme of Zender et al.,
2003 to compute a size-segregated dust flux, which includes entrainment thresholds for saltation,
moisture inhibition, drag partitioning, and saltation feedback. The scheme assumes that vegetation
suppresses dust mobilization by linearly reducing the fraction of bare soil exposed in each grid
cell:

$$A_m = (1 - A_l - A_w)(1 - A_s)(1 - A_V) \tag{1},$$

where $A_l$ is the fraction of land covered by lakes, $A_w$ is the fraction covered by wetlands, $A_s$ is the
fraction covered by snow, and $A_V$ is the fraction covered by vegetation.

For this study, we use VAI as a metric to represent vegetation because it includes not only

leaves but also stems and branches, all of which constrain dust emission. VAI is used to calculate
$A_V$ in equation (1) through

$$A_V = \min \left[1.0, \min(VAI, VAI_t) / VAI_t\right] \tag{2},$$

where $VAI_t$ is the threshold for complete suppression of dust emissions, set here to 0.3 m$^2$ m$^{-2}$
(Zender et al., 2003; Mahowald et al., 1999).

To compute the dust fluxes, we need to convert LAI from LPJ-LMfire to VAI. VAI is

generally defined as the sum of LAI plus stem area index (SAI). Assuming immediate removal of
all dead leaves, the fractional vegetation cover, $\sigma_v$, can be used to represent SAI for the different
PFTs (Zeng et al., 2002). Given that the threshold $VAI_t$ for no dust emission is relatively low (0.3
m$^2$ m$^{-2}$), leaf area dominates stem area in the suppression of dust mobilization in the model. In
areas where LAI is greater than SAI, we therefore assume that SAI does not play a role in
controlling dust emissions, and we set LAI equivalent to VAI. We also assume that C$_3$ and C$_4$
grasses have zero stem area to avoid overestimating VAI during the winter and early spring when
such grasses are dead. Based on the method of Zeng et al., 2002, with modifications, we calculate
VAI in each grid cell as

$$VAI = \max \left( \sum_{PFT=1}^{9} LAI , \sum_{PFT=1}^{7} \sigma_v \right) \qquad (3)$$

where LAI is for the nine PFTs from LPJ-LMfire, and $\sigma_v$ is for just seven PFTs, with $\sigma_v$ for $C_3$
and $C_4$ grasses not considered. Of the nine PFTs, temperate needleleaf evergreen, temperate
broadleaf evergreen, temperate broadleaf summergreen, and $C_3$ grasses dominate the region, with
temperate needleleaf evergreen having the highest LAI in spring. This mix of vegetation type is
consistent with observations (e.g., (McClaran and Van Devender, 1997).
**2.4   Calculation of dust emissions**
Dust emissions are calculated offline in the DEAD dust mobilization module within the
Harvard-NASA Emissions Component (HEMCO). We feed into the DEAD module both the VAI
generated by LPJ-LMfire and meteorological fields from the Modern-Era Retrospective analysis
for Research and Applications (MERRA-2) at a spatial resolution of 0.5° latitude x 0.625°
longitude (Gelaro et al., 2017). Dust emission is nonlinear with surface windspeed. Following
Ridley et al., 2013, we characterize subgrid-scale surface winds as a Weibull probability
distribution, which allows saltation even when the grid-scale wind conditions are below some
specified threshold speed. The scheme assumes that the vertical flux of dust is proportional to the
horizontal saltation flux, which in turn depends on surface friction velocity and the aerodynamic
roughness length $Z_0$. As recommended by Zender et al., 2003, and consistent with Fairlie et al.,
2007 and Ridley et al., 2013, we uniformly set $Z_0$ to 100 μm across all dust candidate grid cells.
With this model setup, we calculate hourly dust emissions for two five-year time slices for
each RCP and condition, covering the present day (2011-2015) and the late-21[st] century (2095-
2099). Dust emissions are generated for four size bins with radii of 0.1 – 1.0 μm, 1.0 – 1.8 μm, 1.8
– 3.0 μm, 3.0 – 6.0 μm. These dust emissions are then applied to GEOS-Chem. Calculated present-
day VAI and fine dust emissions are shown in Figure S3, and we compare modeled VAI with that
observed in Figures S4 and S5.

**2.5 GEOS-Chem**
We use the aerosol-only version of the GEOS-Chem chemical transport model (version
12.0.1; http://acmg.seas.harvard.edu/geos/). For computational efficiency, we apply monthly mean
oxidants archived from a full-chemistry simulation (Park et al., 2004). To isolate the effect of
changing dust mobilization on air quality over the southwestern North America, we use present-
day MERRA-2 reanalysis meteorology from NASA/GMAO (Gelaro et al., 2017) for both the
present-day and future GEOS-Chem simulations. In other words, we neglect the direct effects of
future changes in wind speeds on dust mobilization, allowing us to focus instead on the indirect
effects of changing vegetation on dust. For each time slice, we first carry out a global GEOS-Chem
simulation at 4° latitude x 5° longitude spatial resolution, and then downscale to 0.5° x 0.625° via
grid nesting over the North America domain. In this study, we focus only on dust particles in the
finest size bin (i.e., with radii of 0.1 – 1.0 μm), as these are most deleterious to human health. We
compare modeled fine dust concentrations over southwestern North America for the present-day
against observations from the IMPROVE network in Figures S6-S7.

**3 Results**
**3.1 Spatial shifts in springtime vegetation area index**
Figure 1 shows large changes in the spatial distribution of modeled springtime VAI in the
southwestern North America for the three cases under both RCPs by 2100. In RCP4.5, the
distributions of changes in VAI are similar for the all-factor and fixed-land use cases. Strong
enhancements (up to ~2.5 $m^2$ $m^{-2}$) extend across much of Arizona, especially in the northwestern
corner. The model exhibits moderate VAI increases in most of New Mexico and in the forest
regions along the coast of northwestern Mexico. We find decreases in modeled VAI (up to ~ -1.6
$m^2$ $m^{-2}$) in the southwestern corner of New Mexico, to the east of the coastal forests in Mexico and
in the forest regions near the Mexican border connecting with southern Texas. The similarity
between the all-factor and fixed land use cases indicates the relatively trivial influence of land use
change on vegetation cover in RCP4.5, compared to the effects of climate change and $CO_2$
fertilization. For the fixed-$CO_2$ case, western New Mexico and northern Mexico show greater
decreases in VAI, indicating how $CO_2$ fertilization in the other two cases offsets the effects of the
warmer, drier climate on vegetation in this region. Figure S8 further illustrates the strong positive
impacts that $CO_2$ fertilization has on VAI.

Compared to RCP4.5, the RCP8.5 scenario shows larger changes in climate, $CO_2$

concentrations, and land use by 2100 (Figure 1). The net effects of these changes on vegetation
are complex. As in RCP4.5, Arizona experiences a strong increase in VAI in the all-factor and
fixed-land use cases, but now this increase extends to New Mexico. In contrast to RCP4.5, modeled
VAI decreases in the coastal forest areas in northern Mexico in the all-factor case for RCP8.5. In
the fixed-land use case, however, the VAI decrease in northern Mexico is nearly erased, indicating
the role of vegetation/forest degradation caused by land use practices in this area (Figure S9). For
the fixed-$CO_2$ case for RCP8.5, VAI decreases in nearly all of southwestern North America, except
the northeastern corner of Arizona and the northwestern corner of New Mexico.

To better understand the changes in VAI, we can examine changes in LAI, which

represents the major portion of VAI, for the four dominant plant functional types (PFTs) in this
region. For example, decreases in LAI in the fixed-$CO_2$ case under RCP8.5 are dominated by the
loss of temperate broadleaf evergreen (TeBE) and temperate broadleaf summergreen (TeBS)
(Figure S10). Temperate needleleaf evergreen (TeNE) shows areas of increase in the northern part
and south of Texas in this scenario, while both TeBE and TeBS show increases in northern Arizona
and New Mexico. In other areas, TeBS reveals strong decreases, especially in southern Arizona
and Mexico. As predicted by previous studies (Bestelmeyer et al., 2018; Edwards et al., 2019), $C_3$
perennial grasses ($C_3$gr) in this case decrease across a large swath extending from Arizona through
Mexico, showing the impacts of warmer temperatures and reduced precipitation, as well as (for
Mexico) land use change. Increased fire activity also likely plays a role in the simulated decreases
of forest cover and $C_3$ grasses for RCP8.5 in southern Arizona, where fires together with drought
may have affected landscape succession (Williams et al., 2013; Bodner and Robles, 2017). We
also investigate trends in LAI for different months in spring from the present day to 2100. We find
that the greatest percentage decreases in TeBS and $C_3$ grasses occur in May, consistent with the
largest decreases in precipitation in that month (not shown).
In sum, we find that the warmer and drier conditions of the future climate strongly reduce
vegetation cover by 2100, especially in RCP8.5. In addition, $CO_2$ fertilization and land use
practices further modify future vegetation, but in opposite ways, as illustrated by Figure S8. Under
a warmer climate, higher $CO_2$ concentrations facilitate vegetation growth everywhere in the
southwestern North America, with larger VAI increases occurring over Arizona and New Mexico.
Combined changes in anthropogenic land use – including cropland, pasture, and urban area – are
greater under RCP8.5 than RCP4.5, with large increases in RCP8.5 across Mexico but only modest
changes in Arizona, New Mexico, and Texas (Figure S9). The increases in Mexico result in the
fragmentation of forested landscapes and decrease VAI, especially in coastal forest regions and
along the border with the United States.

**3.2    Spatial variations in spring fine dust emissions**

Unlike the widespread changes in VAI, future changes in fine dust emissions are
concentrated in a few arid areas, including: 1) the border regions connecting Arizona, New Mexico,
and northern Mexico (ANM border), 2) eastern New Mexico, and 3) western Texas (Figure 2). In
RCP4.5, slight increases in fine dust emission (up to ~0.3 kg m$^{-2}$ mon$^{-1}$) are simulated in the ANM
border in all the three cases. In contrast, fine dust emissions decrease by up to ~ -1.0 kg m$^{-2}$ mon$^{-}$
$^1$ in eastern New Mexico and western Texas in RCP4.5 due to warmer temperatures and increasing
VAI. Consistent with the modest changes in VAI (Figure 1), the three cases in RCP4.5 do not
exhibit large differences, with only the fixed-$CO_2$ case showing slightly greater increases in dust
emissions along the ANM border and in western Texas. In RCP8.5 in the all-factor case, spring
fine dust emissions increase slightly by up to ~ 0.4 kg m$^{-2}$ mon$^{-1}$ along the ANM border, but
decrease more strongly in western Texas by up to ~ -1.4 kg m$^{-2}$ mon$^{-1}$ (Figure 2). In contrast, with
fixed $CO_2$ the sign of the change in dust emissions reverses, with significant emissions increases
along the ANM border and in New Mexico. The area with decreasing emissions in western Texas
also shrinks in the fixed $CO_2$ case. These trends occur due to the climate stresses – e.g., warmer
temperatures and decreased precipitation – that impair the growth of temperature broadleaf trees
and $C_3$ grasses. In this case, such stresses are not offset by $CO_2$ fertilization (Figure S10).
Figure 3 shows more vividly the opposing roles of $CO_2$ fertilization and projected land use
change in southwestern North America. In RCP8.5, changing $CO_2$ fertilization alone promotes
vegetation growth and dramatically reduces dust mobilization by up to ~ -1.2 kg m$^{-2}$ mon$^{-1}$. Figure
3 also reveals that land use trends are a major driver of increased dust emissions along the ANM
border and western Texas in RCP8.5,  as crop- and rangelands expand in this region and
temperature broadleaf trees decline (Hurtt et al., 2011). Similarly, the expansion of rangelands in
northern Mexico in RCP8.5 reduces natural vegetation cover there (Hurtt et al., 2011), contributing
to the increase of fine dust emissions by up to ~0.7 kg m$^{-2}$ mon$^{-1}$.

**3.3 Spring fine dust concentrations under the high emission scenario**

Our simulations suggest that fine dust emissions will increase across arid areas in
southwestern North America under RCP8.5, but only if $CO_2$ fertilization is of minimal importance
(Figure 2). To place an upper bound on future concentrations of fine dust in this region, we apply
only the fixed-$CO_2$ emissions to GEOS-Chem at the horizontal resolution of 0.5° x 0.625°. Given
the large uncertainty in the sensitivity of vegetation to changing atmospheric $CO_2$ concentrations
(Smith et al., 2016), we argue that this approach is justified.
Results from GEOS-Chem in the fixed-$CO_2$ case for RCP8.5 show that the concentrations
of spring fine dust are significantly enhanced in the southeastern half of New Mexico and along
the ANM border, with increases up to ~2.5 µg m$^{-3}$ (Figure 4). The model also yields elevated dust
concentrations over nearly the entire extent of our study region by 2100. As Figure 3 implies,
anthropogenic land use along the ANM border contributes to the increased dust emissions in that
area, by up to ~0.7 kg m$^{-2}$ mon$^{-1}$. Climate change impacts on natural vegetation, however, account
for the bulk of the modeled increases in dust emissions in this scenario, by as much as ~1.2 kg m$^{-2}$
$^{2}$ mon$^{-1}$ (Figure 2). The modeled wind fields, which are the same in all scenarios, transport the dust
from source regions, leading to the enhanced concentrations across much of the domain, as seen
in Figure 4. We find that dust concentrations decrease only in a limited area in western Texas due
to decreased pasture (Figures 3 and S9).

**4 Discussion**

We apply a coupled modeling approach to investigate the impact of future changes in

climate, $CO_2$ fertilization, and anthropogenic land use on dust mobilization and fine dust
concentration in southwestern North America by the end of the 21$^{st}$ century. Table 1 summarizes
our findings for the two RCP scenarios and three conditions – all-factor, fixed $CO_2$, and fixed land
use – in spring, when dust concentrations are greatest. We find that in the RCP8.5 fixed-$CO_2$
scenario, in which the effects of $CO_2$ fertilization are neglected, VAI decreases by 26% across the
region due mainly to warmer temperatures and drier conditions, yielding an increase of 58% in
fine dust emission averaged over the southwestern North America. In addition, we find that the
increase in fine dust emission in northern Mexico is mainly driven by the increases in the extent
of cropland and pasture cover in this area, signifying the crucial role of land use practices in
modifying dust mobilization.

Our findings for diminished VAI in the future atmosphere are consistent with observed

trends in vegetation during recent droughts in this region. For example, Breshears et al., 2005
documented large-scale die-off of overstory trees across southwestern North America in 2002-
2003 in response to short-term drought accompanied by bark beetle infestations. Similarly, during
a multi-year (2004-2014) drought in southern Arizona, Bodner and Robles, 2017 found that the
spatial extent of both $C_4$ grass cover and shrub cover decreased in the southeast part of that state.

The 58% increase predicted in this study in fixed-$CO_2$ RCP8.5 scenario is larger than the

26-46% future increases in fine dust for this region predicted by the statistical model of
Achakulwisut et al., 2018. That study relied solely on predictions of future regional-scale
meteorology and did not take into account the change in vegetation, as we do here. In contrast, the
statistical model of Pu and Ginoux, 2017 estimated a 2% decrease in the springtime frequency of
extreme dust events in the Southwest U.S., driven mainly by reductions in bare ground fraction
and wind speed. Like Pu and Ginoux, 2017, we also find that dust emissions decrease across a
broad region of the Southwest when $CO_2$ fertilization is taken into account, as shown in Figure 2.
Pu and Ginoux, 2017 relied on limited data for capturing the sensitivity of dust event frequency to
land cover in this region, and neither that study nor Achakulwisut et al., 2018 considered changes
in land use, as do here. The direct effects of changing wind speed on dust mobilization, however,
are not included in our study, but could be tested in future work.

We further find that consideration of $CO_2$ fertilization can mitigate the effects of changing

climate and land use on dust concentrations in southwestern North America. The all-factor and
fixed-land use simulations both yield decreases of ~20% in mean dust emissions compared to the
early 21$^{st}$ century. In the IPCC projections, $CO_2$ reaches ~550 ppm by 2100 under RCP4.5 and
~1960 ppm under RCP8.5 (Meinshausen et al., 2011). Correspondingly, in the RCP4.5 scenario
for 2100, $CO_2$ fertilization enhances VAI by 30% in the all-factor case compared to the fixed-$CO_2$
case (1.07 $m^2m^{-2}$ vs. 0.79 $m^2m^{-2}$) ; in RCP 8.5, the 2100 enhancement is 64% (1.11 $m^2m^{-2}$ vs. 0.55
$m^2m^{-2}$), as shown in Table 1. These enhancements further decrease fine dust emissions by 21%
under RCP4.5 and 78% under RCP8.5, compared to the present day. Except along the ANM border
and a few other areas, trends in land use have only minor impacts on dust mobilization under the
two RCPs in southwestern North America.

In sum, we find that as atmospheric $CO_2$ levels rise, the effect of enhanced $CO_2$ fertilization

boosts vegetation growth and decreases dust mobilization, offsetting the impacts of warmer
temperatures and reduced rainfall, at least in some areas. These results are consistent with evidence
that enhanced $CO_2$ fertilization is already occurring in arid or semiarid environments like
southwestern North America (Donohue et al., 2013; Haverd et al., 2020). In such environments,
water availability is the dominant constraint on vegetation growth, and the recent enhancement of
atmospheric $CO_2$ may have reduced stomatal conductance and limited evaporative water loss. The
effects of $CO_2$ fertilization on vegetation growth are uncertain, however, and may be attenuated
by the limited supply of nitrogen and phosphorus in soil (Wieder et al., 2015). These nutritional
constraints vary greatly among different PFTs (Shaw et al., 2002; Nadelhoffer et al., 1999).

Understanding the drivers in historic dust trends has sometimes been challenging

(Mahowald and Luo, 2003; Mahowald et al., 2002), making it difficult to validate dust
mobilization models. A further drawback of our approach is that the LPJ-LMfire model is driven
by meteorological fields from just one climate model, GISS-E2-R. Given that the GISS model
yields a conservative prediction of climate change in the southwestern North America compared
to other models (Ahlström et al., 2012; Sheffield et al., 2013), our predictions of the impact of
climate change on dust mobilization may also be conservative. Other uncertainties in our study
can be traced to the dust simulation. The different vegetation types in our model are quantified as
fractions of gridcells, which have relatively large spatial dimensions of ~50 km × 60 km. This
means the model cannot capture the spatial heterogeneity of land cover, and the aerodynamic
sheltering effects of vegetation on wind erosion are neglected, as they are in most 3-D global model
studies. Such sheltering could play a large role in dust mobilization (e.g., Liu et al., 1990). New
methods involving satellite observations of surface albedo promise to improve understanding of
the effects of aerodynamic sheltering on dust mobilization, at least for the present-day (Chappell
and Webb, 2016; Webb and Pierre, 2018). Implementation of aerodynamic sheltering in
simulations of future climate regimes would need to account for fine-scale spatial distributions of
vegetation. In addition, as recommended by Zender et al., 2003, we apply a globally uniform
surface roughness $Z_0$ in the model, which means that the impact of changing vegetation conditions
on friction velocity is not taken into account. Future work could address this weakness by varying
friction velocity according to vegetation type. Finally, our study focuses only on the effect of
changing vegetation on dust mobilization and does not take into account how changing windspeeds
or drier soils in the future atmosphere may more directly influence dust. Given the slight increase
in monthly mean winds in RCP8.5 by 2100, future dust emissions in this scenario could be
underestimated.
Within these limitations, our study quantifies the potential impacts of changing land cover
and land use practices on dust mobilization and fine dust concentration over the coming century
in southwestern North America. Our work builds on previous studies focused on future dust in this
region by (1) more accurately capturing the transport of dust from source regions with a dynamical
3-D model, (2) considering results with and without $CO_2$ enhancement, and (3) including the
impact of land use trends. Given the many uncertainties, it is challenging to gauge which of the
three factors investigated here – climate impacts on vegetation, $CO_2$ fertilization, or land use
change – will play the dominant role in driving future changes in dust emissions and concentrations.
This study thus brackets a range of possible dust scenarios for the southwestern North America,
with the simulation without $CO_2$ fertilization placing an upper bound on dust emissions. In the
absence of increased $CO_2$ fertilization, our work suggests that vegetated area will contract in
response to the warmer, drier climate, exposing bare land and significantly increasing dust
concentrations by 2100.
Dust enhancement could thus impose a potentially large climate penalty on $PM_{2.5}$ air
quality, with consequences for human health across much of southwestern North America, where
much of the current population is of Native American and/or Latino descent. In New Mexico for
example, 10% of the population is Native American and 50% identifies as either Hispanic or Latino.
By some measures, New Mexico has also one of highest poverty rates of the United States
(https://www.census.gov/quickfacts/NM, last accessed on August 20, 2020). In this way, our
finding of the potential for an increased dust burden in the future atmosphere has special relevance
for environmental justice in this region.

**Code and data availability**
GEOS-Chem model codes can be obtained at http://acmg.seas.harvard.edu/geos. LPJ-LMfire
model codes can be obtained at https://github.com/ARVE-Research/LPJ-LMfire. IMPROVE
datasets are available online at http://vista.cira.colostate.edu/improve. Any additional information
related to this paper may be requested from the authors.

**Author contributions**
Y.L. conceived and designed the study, performed the GEOS-Chem simulations, analyzed the data,
and wrote the manuscript, with contributions from all coauthors. J.O.K. performed the LPJ-LMfire
simulations.

**Competing interests**
The authors declare that they have no competing interest.

**Acknowledgments**


This research was developed under Assistance Agreements 83587501 and 83587201 awarded by
the U.S. Environmental Protection Agency (EPA). It has not been formally reviewed by the EPA.
The views expressed in this document are solely those of the authors and do not necessarily reflect
those of the EPA. We thank all of the data providers of the datasets used in this study. PM data
was provided by the Interagency Monitoring of Protected Visual Environments (IMPROVE;
available online at http://vista.cira.colostate.edu/improve). IMPROVE is a collaborative
association of state, tribal, and federal agencies, and international partners. U.S. Environmental
Protection Agency is the primary funding source, with contracting and research support from the
National Park Service. JOK is grateful for access to computing resources provided by the School
of Geography and the Environment, University of Oxford. The Air Quality Group at the University
of California, Davis is the central analytical laboratory, with ion analysis provided by the Research
Triangle Institute, and carbon analysis provided by the Desert Research Institute. We acknowledge
the World Climate Research Programme's Working Group on Coupled Modelling, which is
responsible for CMIP, and we thank the group of NASA Goddard Institute for Space Studies for
producing and making available their GISS-E2-R climate model output. For CMIP the U.S.
Department of Energy's Program for Climate Model Diagnosis and Intercomparison provides
coordinating support and led development of software infrastructure in partnership with the Global
Organization for Earth System Science Portals. The GISS-E2-R dataset were downloaded from
https://cmip.llnl.gov/cmip5/. We thank the Land-use Harmonization team for producing the
harmonized set of land-use scenarios and making available the dataset online at
http://tntcat.iiasa.ac.at/RcpDb/. We also thank the founder, organizers, and participants of the
Degree Confluence Project (www.confluence.org).

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

**Figures and tables**

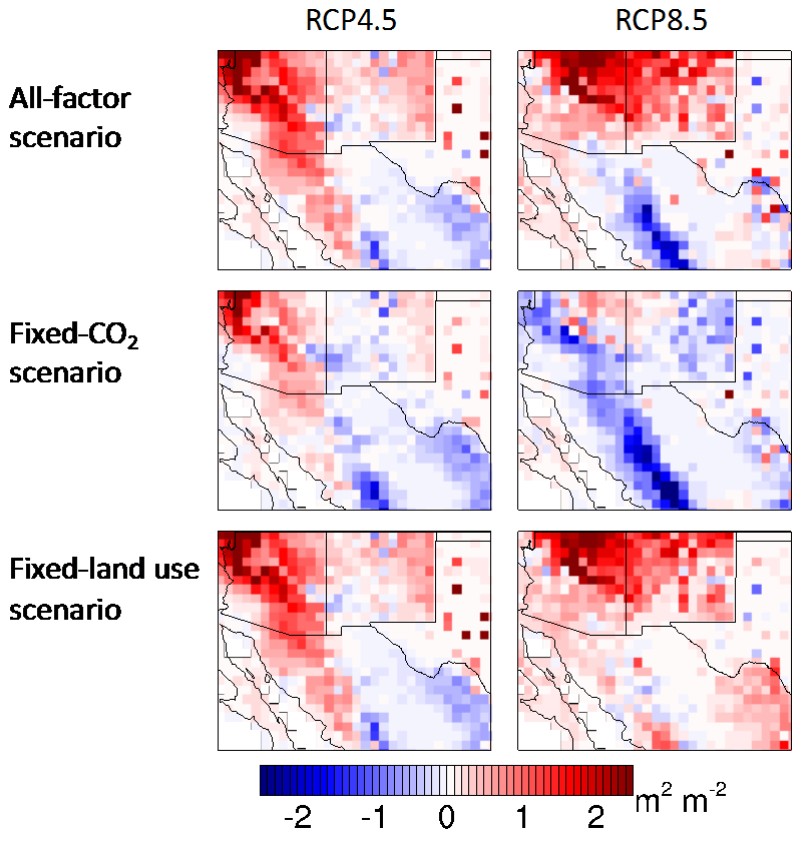


**Figure 1.** Simulated changes in spring averaged monthly mean vegetation area index (VAI) in
southwestern North America under the three conditions for RCP4.5 and RCP8.5. Changes are
between the present day and 2100, with five years representing each time period. The All-factor
case (top row) includes the effects of climate, $CO_2$ fertilization, and anthropogenic land use on
vegetation. Only climate and land use are considered in the Fixed-$CO_2$ case (middle), and only
climate and $CO_2$ fertilization are considered in the Fixed-land use case (bottom). Results are from
LPJ-LMfire.

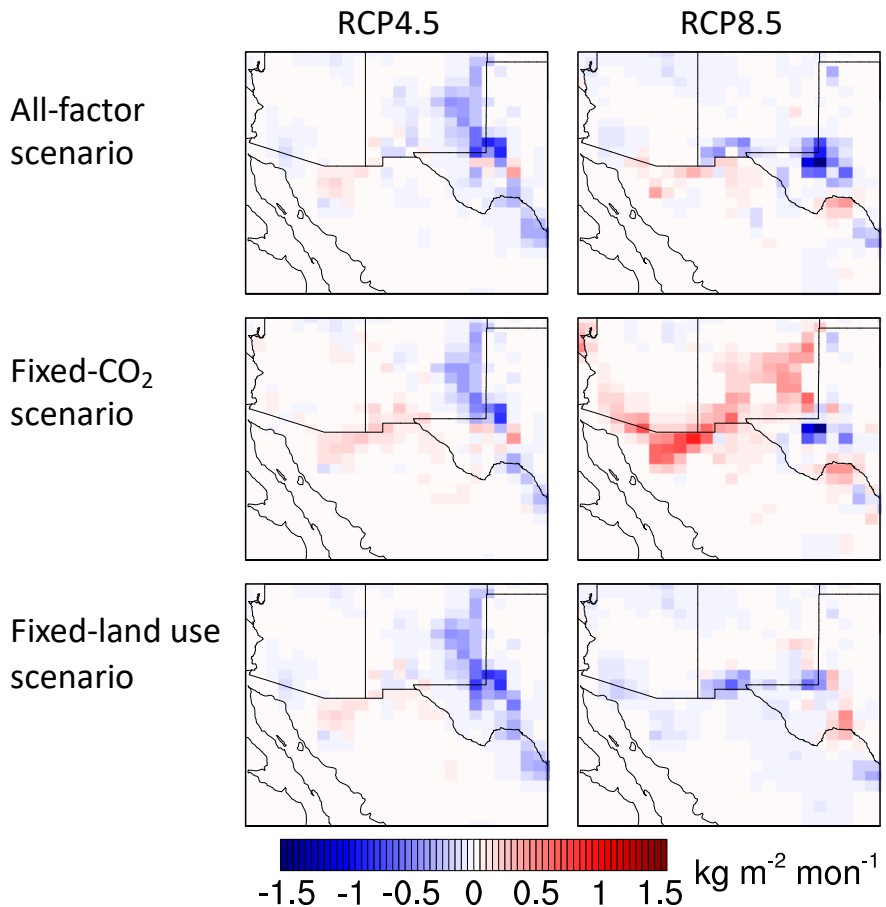

**Figure 2.** Simulated changes in spring averaged monthly mean dust emission in southwestern North America under the three conditions for RCP4.5 and RCP8.5. Changes are between the present day and 2100, with five years representing each time period. The top row shows results for the all-factor condition, the middle row is for the fixed-$CO_2$ condition, and the bottom row is for the fixed-land use condition. Cases are as described in Figure 1. Results are generated offline using the GEOS-Chem emission component (HEMCO).

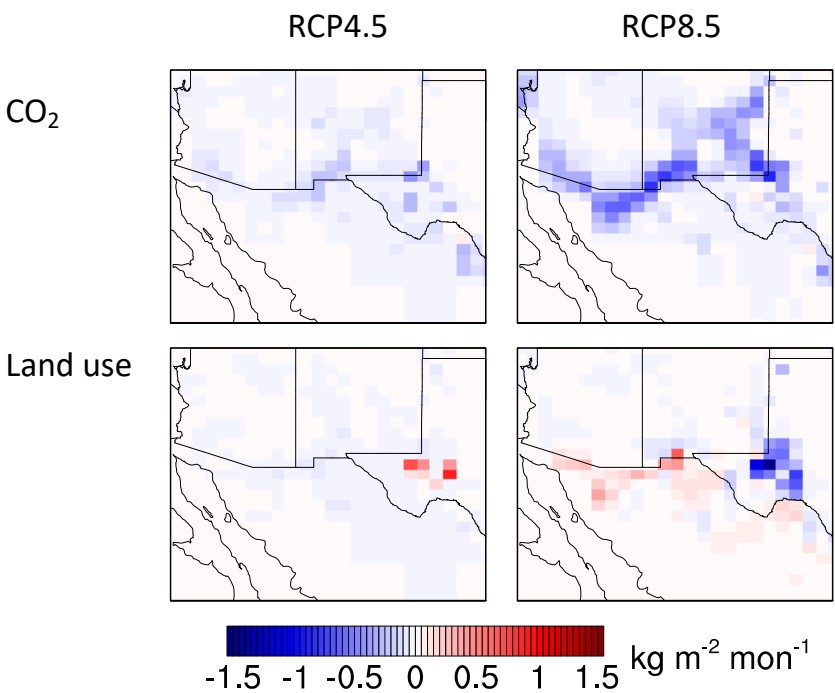

710

**Figure 3.** Contributions of $CO_2$ fertilization and land use change to changing dust emissions in spring in southwestern North America for RCP4.5 and RCP8.5. Changes are between the present day and 2100, with five years representing each time period. The top row shows the response of dust emission to only $CO_2$ fertilization and the bottom row shows the response to only trends in land use. Results are generated offline using the GEOS-Chem emission component (HEMCO).

716

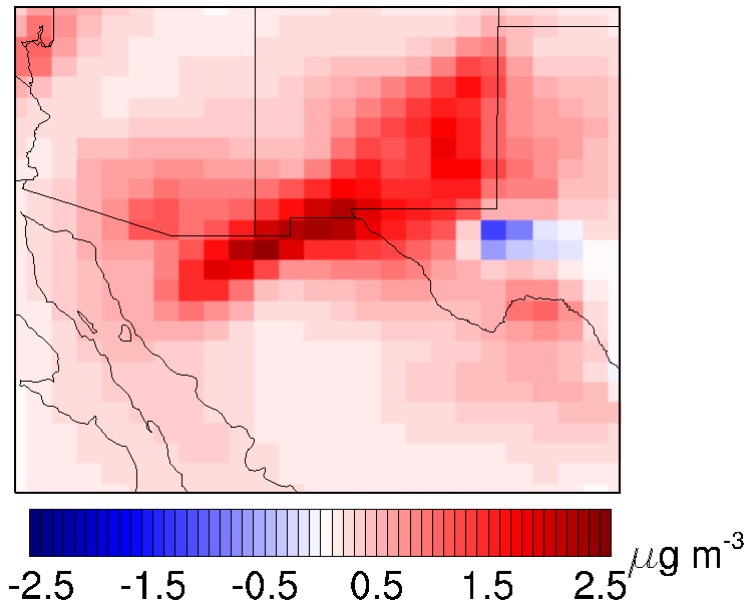

717

**Figure 4.** Simulated changes in springtime mean concentrations of fine dust over southwestern

North America for the RCP8.5 fixed-$CO_2$ case, in which the effects of $CO_2$ fertilization are

neglected. Changes are between the present day and 2100, with five years representing each time

period. Results are from GEOS-Chem simulations at 0.5° x 0.625° resolution.


**Table 1.** Averaged spring vegetation area index (VAI) and fine dust emission in southwestern
North America for the present-day and future for two scenarios (RCP4.5 and RCP8.5) and three
cases. The all-factor case includes changes in climate, land use, and $CO_2$ fertilization; the fixed-
$CO_2$ case includes changes in only climate and land use; and the fixed-land use case includes
changes in only climate and $CO_2$. The rows labeled "2100-2010, %" give the percentage changes
in VAI and fine dust emissions between the present day and future, with positive values denoting
increases in the future.

| | | VAI[b], $m^2$ $m^{-2}$ | | | Fine dust emission[b], kg $m^{-2}$ $mon^{-1}$ | | |
|---|---|---|---|---|---|---|---|
| | | All-factor | Fixed $CO_2$ | Fixed land use | All-factor | Fixed $CO_2$ | Fixed land use |
| **RCP4.5** | **2010[a]** | 0.75±0.26 | 0.71±0.24 | 0.75±0.26 | 0.10±0.07 | 0.11±0.08 | 0.10±0.07 |
| | **2100[a]** | 1.07±0.48 | 0.79±0.34 | 1.07±0.48 | 0.08±0.04 | 0.10±0.05 | 0.08±0.04 |
| **2100-2010, %** | | **42** | **12** | **42** | **-25** | **-4** | **-26** |
| **RCP8.5** | **2010[a]** | 0.80±0.27 | 0.75±0.24 | 0.75±0.24 | 0.09±0.04 | 0.09±0.05 | 0.09±0.04 |
| | **2100[a]** | 1.11±0.71 | 0.55±0.33 | 0.55±0.33 | 0.07±0.04 | 0.14±0.09 | 0.07±0.06 |
| **2100-2010, %** | | **38** | **-26** | **52** | **-20** | **58** | **-16** |

[a]Each time slice represents 5 years (i.e., 2011-2015 represents the 2010 time slice and 2095-2099 represents the 2100
time slice); [b]Values are spring (MAM) averages over southwestern North America.