# Peer review of "Response of dust emissions in southwestern North America to 21st century trends in climate, CO$_2$ fertilization, and land use: Implications for air quality"

_Atmospheric Chemistry and Physics, 2020_

## Referee Comment (RC1) · Anonymous Referee #1 · 12 May 2020

This manuscript describes a coupled modeling study to investigate the role of climate change on dust emissions in southwestern North America (SW). The role of dust emissions and transport in the SW is important for air quality impacts in the region and has been suggested by other research that dust concentrations and associated impacts will likely worsen. The authors incorporate a dynamic vegetation model and a chemical transport model with two different future emission scenarios to examine the effects of land use change and CO2 fertilization on dust emissions in the SW. They found that under the most extreme future warming scenario used (RCP8.5), the absence of

[Figure]

CO2 fertilization provides an upper bound on increased dust emissions across the SW, but especially in SE New Mexico (NM) and the border between NM and Mexico. It is important to consider various causal impacts in order to design appropriate mitigation strategies, so the types of analyses described in this paper are important and worthwhile. However, a significant weakness of the paper is the discussion and accounting for the role of drought impacts on dust emissions. The authors don't reference this very important impact to the region and how it might impact CO2 fertilization and the competing impacts on plant growth through water stress. The paper would also benefit from additional organization and clarification. I recommend a major revision to deal with some of these issues- see detailed in the comments below.

Line 14: How is surface air quality defined here? Do the authors mean only particulate matter?

Line 16: Perhaps refer to the "spring time" earlier, it's not clear whether decreasing trends were observed year round and then increasing trends were only in spring? (how is spring defined?)

Line 20: Perhaps refer to the fact that only these two drivers were investigated- the role of drought is very important in this region and does not seem to be addressed in this work. (e.g., see Archer and Predick, 2008; MacDonald, 2010; Prein et al., 2016; Stahle, 2020; William et al., 2020)

Line 22: Instead of RCP8.5, just use "most extreme future warming scenario" like was used in line 15. Or define RCP8.5.

Line 24: Above some reference value?

Line 25: It would be helpful to the reader if the authors leave them with a motivation for this study. Why should the reader care? Future mitigation strategies? Also, some reference to the fact that drought was not studied because many readers will be familiar with the role of drought in this area and wonder if/how/why it was accounted for in this

study.

Line 32: And soil moisture? Drought? (for example, see references listed above)

Line 39: Here and throughout the paper it is unclear what the authors define as "climate change" and it is important to define it here. Do they just mean increased emissions? Or increased temperatures? Increased drought?

Line 39-42: These sentences read like they should come towards the end of the Introduction.

Line 44-49: Where did these studies occur? It also seems that the estimates would depend largely on the particular region of study, since different regions may have different controlling factors. Line 60: This study indicates the importance of drought, but again, it is not clear whether the impacts of increased drought is included in this study?

Line 69: If "climate-induced changes" includes the role of drought, it should be described here because it is unclear. If it is not, the authors need to address why this very important role was not considered.

Line 75: Like the abstract, the end of the Introduction would benefit from an implications statement, or some description of what these results could inform in terms of public policy or future studies.

Line 77: An overall statement about the Methods section. It is difficult to follow, descriptions of some of the models and methods are scattered throughout the sections. The entire section would benefit from a streamlining and overall organization. It seems like the authors are giving a brief overview of the method at the beginning, which is fine, but in its present form it includes some details that leave the reader looking around for descriptions that aren't included until later. Perhaps leave the overview very general and then describe each step in more detail. For example:

Line 85:86- Time periods aren't given, GISS is discussed here but then again in line 133 (maybe a separate "GISS" section, like the other models have?).

Line 88: Again, what does "changes in climate" mean in this context?

Line 93: How is "fine dust" defined (again, this comes later).

Line 100: We are in the Methods section?

Line 102-107: This seems misplaced, perhaps it should go in a separate "GISS" section?

Line 109: What land use fields are included in the model and where do they come from? Some reference to this is included in line 237 but would be useful to know sooner. Where does the vegetation information come from? Is it representative of desert vegetation? Where does wildfire information come from and does it change over time? Do the meteorological anomalies characterize future drought?

Line 118: "Future land use scenarios applied follow CMIP5". Can the authors expand and define CMIP5? What all types of land use scenarios are included?

Line 121-122: Some discussion here regarding how the model accounts for hydrologic feedbacks, such as whether plants react to water limitation?

Line 122: "...and analyze results over..." This sentence is redundant and unnecessary.

Line 125-128: Discussion of RCP4.5 and RCP4.8 seems out of order here.

Line 129-133: Redundant, see lines 85-87. Again, move the GISS information into a GISS section.

Line 161: How representative are these of desert plants in the Southwest?

Line 165: I assume (based on equation 3) that 7 different PFTs are included to represent stem area index? What are they?

Line 170: Are all plants represented here responsive to $CO_2$ fertilization? How do the effects of drought, heat, and evapotranspiration offset gains in $CO_2$ fertilization and can this be captured by the model? If not, it should be stated.

Line 177: MERRA is mentioned here for the first time?

Line 202: Define "springtime"

Line 205: These boundaries are not shown on the figures and probably aren't important to mention here.

Line 237: This description of land use change would be helpful earlier.

Line 246: How is "desertification" defined? Does this imply anything about drought?

Line 257: How are "climate stresses" defined and quantified in the model? This implies impacts from drought and water stress on plants, but as mentioned before, this doesn't seem to be captured by the model? Should "temperature" be "temperate"?

Line 264: What is the land use type shifting towards in these regions?

Line 277-278. I am not sure I understand this sentence. Land use is the driver, but climate change makes up the bulk of the increases?

Line 279: The authors seem to be implying that winds are also involved in these differences?

Line 292: This wasn't specifically shown in the results (shifts in land surface type).

Line 298-299: And this study doesn't include changes in wind speed, so it's hard to say that the differences between the Pu and Ginoux study are primarily due to the changes in vegetation.

Line 308: So that I am understanding what is presented in the Table, $CO_2$ fertilization would correspond to "fixed land use" but I don't see 30% or 64% in the table?

Line 312-213: But, as stated previously, it is unclear whether future drought is accounted for, or whether the role of increased temperature and water stress on whether plants are responsive to $CO_2$ fertilization is addressed. This seems like an important question the authors need to address, as it could change the directions of trends in

dust emission. The authors need to discuss how or whether this was accounted for.

Line 367: References: There appears to be formatting inconsistencies with several of the references. I encourage the authors to check their reference manager settings (e.g., line 396, 399, 417, 433, 435, etc.). In addition, "doi's" were not included for any of the references.

Line 486: Figure 1: This is the first time land use is referred to as "anthropogenic" and would benefit from a description of what this means (in text).

Line 517: In the "a" description, include whether "2010" is the first year in the 5 year slice.

Archer and Predick, 2008, "Climate change and ecosystems of the Southwestern United States", Rangelands, 30(3):23-28

MacDonald, G.M., 2010 "Water, climate change, and sustainability in the Southwest", PNAS, 107(50).

Prein et al., 2016, "Running dry: The U.S. Southwest's drift into a drier climate state", GRL, 43, doi:10.1002/2015GL066727.

Stahle, D.W. 2020, "Anthropogenic megadrought", Science, 368 (6488).

Williams, A. P., et al., 2020, "Large contribution from anthropogenic warming to an emerging North American megadrought", Science, 368 (314-318).
* * *

---

## Referee Comment (RC2) · Anonymous Referee #2 · 27 May 2020

The authors present a study of how dust emissions across southwestern US states could respond to projected climate changes, elevated atmospheric CO2 and land use change. Projected climate changes are assessed for two Representative Concentration Pathways (RCP 4.5 and 8.5) representing moderate and continued increases in greenhouse gas concentrations through the 21st century. The effects of the climate projections on surface erodibility are represented through a dynamic vegetation model that is linked to a dust emission scheme and the GEOS-Chem chemical transport model. The general subject matter of the manuscript and approach taken is consistent

with regional dust modelling approaches today. Linking a dynamic vegetation model to a dust model to investigate projected climate changes is novel, not straightforward, and has potential to provide new insights into the effects (and interactions) of dust emission under changing land uses and climate.

Overall, my assessment is that, while the subject matter is timely, the manuscript has a number of shortcomings that reduce the relevance of the work and confidence that the conclusions are adequately supported by the approach. These include:

1) While the first paragraph of the Introduction seeks to establish the relevance of the study, this is done only at a very high level and specific research and management impetus are not provided. This high-level treatment of the rationale for the work is carried throughout the manuscript, with the text rarely going deeper than general drivers and responses to justify why the work is important, how it can have impact, who it may have impact for, or how any of the processes and interactions between vegetation, land use and climate actually work and may influence future dust emissions. The superficial treatment of these important elements reduces the impact of the work. Adding detail to these elements would give the work more weight and enable the authors to show exactly what the new insights are that they provide, how they are relevant, and where key uncertainties are.

2) A focus of the manuscript is establishing how future vegetation and land use changes may influence dust emissions. However, the authors have not grounded the manuscript in the present situation – What types of vegetation communities are there across the study area? What types of land use changes are occurring today? How important is land use versus land management? How do these present changes relate to the modeled vegetation and land use change scenarios? How are the vegetation communities changing today? What are the implications of vegetation change trajectories today for future responses to elevated CO2, climate change, and land use? How are these changes related to and influence aeolian processes? By not addressing these questions, the work presents as a typical dust modelling study and/but detached from reality.

Expanding the Introduction and Discussion sections is needed to ground the work 'in the real world' and could help the authors demonstrate the relevance and contribution of the study (point #1 above).

3) The modeled vegetation changes appear unconnected to vegetation changes occurring across southwestern US landscapes today and are not adequately represented in the dust model. As described in Sections 2.2 and 2.3, the DEAD model is used to estimate dust emissions, with vegetation effects represented through a linear adjustment term Av that is calculated from VAI that is the sum of leaf and stem area indices. This approach makes two assumptions that are inconsistent with the physics of aeolian transport and drag partition theory: 1) fractional vegetation cover adequately represents lateral surface aerodynamic sheltering – ergo structural changes in surface roughness due to changing vegetation were not represented while they are likely to have a greater influence on dust emissions than fractional ground cover (Av), and 2) adjustments to the fractional vegetation cover can be made through a dynamic vegetation model (to represent vegetation change) that are separate to the dust model drag partition scheme and its use of aerodynamic roughness lengths (z0) – creating a functional disparity in how vegetation is represented in different parts of the model. I identify these issues in full recognition of the difficulty of accurately representing future vegetation change in a dust model. However, these two assumptions also potentially undermine the validity of the model experiments and so need to be addressed transparently. Further, what are the implications of the model parameterization for the rigor of the results? How much confidence can we have in the outcomes of the study? Where are the gaps that need to be addressed? Turning this challenge into a positive – what insights does this work provide for how future research can address interactions among climate change, vegetation change, land use and dust emissions?

4) Literature cited is constrained to dust modelling studies and a few supporting studies related to the vegetation and climate modelling. In addressing my concerns above, the authors could draw on the rich and diverse literature addressing vegetation and land

use changes, and their interactions with aeolian processes, across the southwestern US.

Some specific concerns are as follows:

Line 65: Given the focus of the manuscript on land use and vegetation change as a driver of changing dust emissions, the introduction would benefit from inclusion of a review paragraph/synthesis of the types of vegetation and the trajectories of these ecosystems across the southwest today. This is likely to have important implications for trends in dustiness, with pervasive vegetation changes influencing surface aerodynamics and wind erosivity. The authors might also comment on the likely sensitivity of these vegetation communities to elevated $CO_2$. See for example references within:

Bestelmeyer et al., 2018. The Grassland-Shrubland Regime Shift in the Southwestern United States: Misconceptions and Their Implications for Management. Bioscience 68, 678-690.

Edwards et al., 2019. Climate change impacts on wind and water erosion on US rangelands. Journal of Soil and Water Conservation. Vol. 74, 405-418. doi:10.2489/jswc.74.4.405.

Line 110: How important is fire in the study area, if at all for the changes under investigation? Supporting references would help.

Line 125: It would be helpful if the authors can define what they mean by vegetation structure. Is this purely geometric (e.g., height, width of plants), or does this include spatial patterns in landscapes?

Line 157: The authors use an estimate of fractional vegetation cover to linearly account for vegetation effects which are predominantly lateral and non-linear for saltation flux and dust emission. While working within the constraints of the DEAD model, the authors should recognize the limitations of this approach and implications for the sensitivity of the model to vegetation change and accuracy of its representation of dust

emission responses.

Line 161: How representative are these classes of vegetation communities across the southwest? How do they relate to actual patterns of vegetation? For reference, the authors might look at NRCS ecological site descriptions across the study area.

Line 166: Although, during the first half of spring in the desert southwest, C3 shrubs (e.g., Prosopis glandulosa) may not have leaves such that the main aerodynamic effect is provided by branches and stems. It would be instructive to link actual plant phenology in the study area to what is/is not represented in the vegetation model.

Line 174: How did the authors parameterize the drag partition scheme and represent land use change effects in the dust model? In DEAD, these are represented through the MB95 drag partition scheme, with aerodynamic roughness lengths (z0) assigned to land cover classes. As dust emission is a lateral process, z0 and the drag partition should have a larger effect on dust emission than fractional cover via VAI. If z0 was not changed consistently with the fractional cover of vegetation, the model would represent an inconsistent vegetation effect and would likely not capture the nature of dust emission responses to the examined scenarios.

Line 180: Do the authors mean saltation, or dust emission? Although a general term, dust shouldn't be saltating.

Line 192: Can the authors describe the implications of not changing wind speed? Would you anticipate wind speed changes in response to regional vegetation (roughness) change and changes in synoptic meteorology?

Line 201: Discussion point - what about changes in seasonality due to changes in plant phenological changes due to species change and change in the timing of warming and precipitation? This is partially addressed in the results, but would benefit from further discussion linked to actual plant communities.

Line 232: How do these modeled changes relate to the vegetation communities in

these locations in reality and their dynamics? Line 235: The effect of vegetation on dust emission shouldn't be reduced to growth as it is the kinds and proportions of vegetation in the landscape that influence surface aerodynamic roughness and spatial patterns of dust emission. These changes aren't represented in the model, but do need to be addressed by the authors.

Line 246: Can the authors define what they mean by desertification, and how this differs to the vegetation changes (grass-shrub transitions) that have already occurred over much of this region? e.g., for reference see Bestelmeyer, B.T., Okin, G.S., Duniway, M.C., Archer, S.R., Sayre, N.F., Williamson, J.C., Herrick, J.E., 2015. Desertification, land use, and the transformation of global drylands. Frontiers in Ecology and the Environment 13, 28-36.

Line 269: What conditions would make CO2 of limited importance? Can the authors explain and expand on this in the Discussion? Will CO2 be the main driver of vegetation change, or are other factors likely to be more important/have been important in the past that are likely to influence future trends? (e.g., vegetation state transitions driven in part by land management, not just land use)

Line 278: It would help for the authors to expand on this point about wind as my understanding is that wind speeds were not adjusted for climate changes in the scenarios/simulations.

Line 280: Again, it would be good if the authors can be specific about both vegetation change and land use change. For example, what is the changing land use in west Texas in this scenario?

Line 298: I agree with this statement about the importance of robust representation of both future vegetation changes and the sensitivity of dust emissions to these changes. However, I question whether this need has actually been addressed in the present study. See my major concerns above relating to: 1) description of changes lacking detail and grounding in actual vegetation and land use changes occurring across the

southwest, and 2) physical representation of vegetation in the dust model ignores the major effect of vegetation on dust emission (lateral process) and the interactions with vegetation changes that are likely to occur.

Line 312: I think the emphasis on CO2 perhaps oversimplifies the controls. These dryland systems are largely water, not nutrient, limited. But not only cover - this will also be C3 vs C4 dominance and so the proportions and kinds of vegetation on these landscapes will influence responses to elevated CO2. Vegetation state changes today and into the future (influenced to some degree by CO2) are likely to have a far greater effect on the structure and cover of protective roughness.

---

## Author Response (AR1)

**Response to reviewers**

We thank the reviewers for their insightful comments. Below we provide detailed responses in black, with quotation marks showing the changes made in the manuscript. The reviewers' comments are in blue, and line numbers in blue refer to the original submission. The line numbers in black refer to the revised manuscript.

**Summary of revisions.** The manuscript now goes into more detail about the processes simulated by LPJ-LMfire, and we more clearly acknowledge the shortcomings in our dust simulation. We have attempted to show how our modeled trends in vegetation across the southwestern United States are consistent with present-day changes. We also now emphasize that the meteorological conditions projected for 2100 in our model are consistent with the increased drought predicted by other studies. We clarify that the trends in vegetation and hence dust are caused by three factors – changes in climate (temperature and precipitation), enhanced  $CO_2$  fertilization, and land use change.

Finally, we have reorganized parts of the Introduction and Methods sections. We now use the term "scenario" to refer to the IPCC scenarios RCP4.5 and RCP8.5, and we use the term "case" to refer to the three conditions applied (all-factor, fixed CO2 fertilization, and fixed land use.).

**Author Response to Reviewer #1**

This manuscript describes a coupled modeling study to investigate the role of climate change on dust emissions in southwestern North America (SW). The role of dust emissions and transport in the SW is important for air quality impacts in the region and has been suggested by other research that dust concentrations and associated impacts will likely worsen. The authors incorporate a dynamic vegetation model and a chemical transport model with two different future emission scenarios to examine the effects of land use change and CO2 fertilization on dust emissions in the SW. They found that under the most extreme future warming scenario used (RCP8.5), the absence of CO2 fertilization provides an upper bound on increased dust emissions across the SW, but especially in SE New Mexico (NM) and the border between NM and Mexico. It is important to consider various causal impacts in order to design appropriate mitigation strategies, so the types of analyses described in this paper are important and worthwhile.

However, a significant weakness of the paper is the discussion and accounting for the role of drought impacts on dust emissions. The authors don't reference this very important impact to the region and how it might impact CO2 fertilization and the competing impacts on plant growth through water stress. The paper would also benefit from additional organization and clarification. I recommend a major revision to deal with some of these issues- see detailed in the comments below.

The reviewer raises an important issue. It is true that our study does not consider the direct effects of changes in meteorology - e.g., changes in wind speeds - on dust emissions. However, we do take into account the effects of soil moisture and drought on plant growth, and such effects do, in turn, influence dust mobilization. Put another way, our study considers the impact of future drought

on vegetation, and from there on dust mobilization. We now clarify this point in several places of our manuscript.

Lines 13-29: "Here we drive a dynamic vegetation model (LPJ-LMfire) with future scenarios of climate and land use, and link the results to a chemical transport model (GEOS-Chem) to assess the impacts of land cover on dust mobilization and fine dust concentrations (defined as dust particles less than 2.5 microns in diameter) on surface air quality. In the most extreme warming scenario (RCP8.5), we find that surface temperatures in southwestern North America during the season of greatest dust emissions (March, April, and May) warm by 3.3 K and precipitation decreases by nearly 40% by 2100. These conditions lead to vegetation dieback and an increase in dust-producing bare ground. Enhanced CO2 fertilization, however, offsets the modeled effects of warming temperatures and rainfall deficit on vegetation in some areas of the southwestern United States. Considering all three factors in RCP8.5 scenario, dust concentrations decrease over Arizona and New Mexico in spring by the late 21st century due to greater CO2 fertilization and a more densely vegetated environment, which inhibits dust mobilization. Along Mexico's northern border, dust concentrations increase as a result of land use intensification. In contrast, when CO2 fertilization is not considered in the RCP8.5 scenario, vegetation cover declines significantly across most of the domain by 2100, leading to widespread increases in fine dust concentrations, especially in southeastern New Mexico (up to  $\sim 2.0 \ \mu g \ m^{-3}$  relative to the present day) and along the border between New Mexico and Mexico (up to  $\sim 2.5 \ \mu g \ m^{-3}$ )."

Lines 43-48. "Wind speed and vegetation cover are two key factors that determine soil erodibility and dust emissions. Wind gusts mobilize dust particles from the earth's surface, while vegetation constrains dust emissions by reducing bare land extent and preserving soil moisture (Zender et al., 2003). The high temperatures and reduced soil moisture characteristic of drought play an important role in dust mobilization, since the resulting loss of vegetative cover increases soil erosion (Archer and Predick, 2008; Bestelmeyer et al., 2018).

Lines 56-60: "Climate models predict a warmer and drier environment in southwestern North America through the 21st century, with more frequent and severe drought (Seager and Vecchi, 2010; MacDonald, 2010; Stahle, 2020; Prein et al., 2016; Williams et al., 2020). Such conditions would decrease vegetative cover and allow for greater dust mobilization."

Lines 152-157: "In our study, we do not specifically track drought frequency under future climate, as the definition of drought is elusive (Andreadis et al., 2005; van Loon et al., 2016). Nonetheless, the meteorological conditions predicted in the RCP8.5 scenario for 2100 align with previous studies projecting increased risk of drought in this region (e.g., Williams et al., 2020), and as we shall see, such conditions, in the absence of  $CO_2$  fertilization, result in decreased vegetation and greater dust mobilization."

In the Discussion section, we clarify a limitation of our study.

Lines 445-447: "Finally, our study focuses only on the effect of changing vegetation on dust mobilization and does not take into account how changing windspeeds or drier soils in the future atmosphere may more directly influence dust."

Finally, LPJ-LMfire takes into account the interactions of water stress and CO2 fertilization.

Lines 59-62: "On the other hand, elevated CO2 concentrations could increase photosynthesis and decrease transpiration of some vegetation species, allowing for more efficient water use and enhancing growth (Poorter and Perez-Soba, 2002; Polley et al., 2013)."

Lines 167-170: "The model considers the coupling of different ecosystem processes, such as the interactions between  $CO_2$  fertilization, evapotranspiration, and temperature, as well as the competition among different PFTs for water resources (e.g., precipitation, surface runoff, and drainage)."

Line 14: How is surface air quality defined here? Do the authors mean only particulate matter?

Yes. We have restated as follows.

Lines 14-16: "..., and link the results to a chemical transport model (GEOS-Chem) to assess the impacts of land cover on dust mobilization and fine dust concentrations (defined as dust particles less than 2.5 microns in diameter) on surface air quality."

Line 16: Perhaps refer to the "spring time" earlier, it's not clear whether decreasing trends were observed year round and then increasing trends were only in spring? (how is spring defined?)

Done.

Lines 17-19: "In the most extreme warming scenario (RCP8.5), we find that surface temperatures in southwestern North America during the season of greatest dust emissions (March, April, and May) warm by 3.3 K and precipitation decreases by nearly 40% by 2100."

Line 20: Perhaps refer to the fact that only these two drivers were investigated- the role of drought is very important in this region and does not seem to be addressed in this work. (e.g., see Archer and Predick, 2008; MacDonald, 2010; Prein et al., 2016; Stahle, 2020; William et al., 2020)

In this paper, we consider three drivers of dust: climate (including drought), CO2 fertilization of vegetation, and land use. We have clarified this issue in the Abstract, as described above, and elsewhere.

Lines 39-42: "In this study, we use a suite of models to predict the future influence of three factors – climate change, increasing atmospheric  $CO_2$  concentrations, and land use change – on land cover in this region, and assess the consequences for dust mobilization and dust concentrations."

Lines 375-382: "We apply a coupled modeling approach to investigate the impact of future changes in climate,  $CO_2$  fertilization, and land use on dust mobilization and fine dust concentration in southwestern North America by the end of the 21st century. Table 1 summarizes our findings for the two RCP scenarios and three conditions – all-factor, fixed CO2, and fixed land use – in spring, when dust concentrations are greatest. We find that in the RCP8.5 fixed-CO2 scenario, in which the effects of CO2 fertilization are neglected, VAI decreases by 26% across the region due

mainly to warmer temperatures and drier conditions, yielding an increase of 58% in fine dust emission averaged over the southwestern North America."

We have also added the citations suggested by the reviewer.

Line 22: Instead of RCP8.5, just use "most extreme future warming scenario" like was used in line 15. Or define RCP8.5.

Fixed.

Line 17-19: "In the most extreme warming scenario (RCP8.5), we find that surface temperatures in southwestern North America during the season of greatest dust emissions (March, April, and May) warm by 3.3 K and precipitation decreases by nearly 40% by 2100."

Line 24: Above some reference value?

We have clarified as follows.

Lines 25-29: "In contrast, when CO2 fertilization is not considered in the RCP8.5 scenario, vegetation cover declines significantly across most of the domain by 2100, leading to widespread increases in fine dust concentrations, especially in southeastern New Mexico (up to ~2.0  $\mu$ g m-3 relative to the present day) and along the border between New Mexico and Mexico (up to ~2.5  $\mu$ g m-3)."

Line 25: It would be helpful to the reader if the authors leave them with a motivation for this study. Why should the reader care? Future mitigation strategies? Also, some reference to the fact that drought was not studied because many readers will be familiar with the role of drought in this area and wonder if/how/why it was accounted for in this study.

We have made the following changes.

Lines 11-13: "The consequences of climate change for dust mobilization and concentrations are unknown, but could have large implications for human health, given connections between dust inhalation and disease."

Lines 29-31: "Our results have implications for human health, especially for the health of the indigenous people who make up a large percentage of the population in this region."

Lines 103-105: "Given the deleterious impacts of airborne dust on human health, our dust projections under different climate scenarios have value for understanding the full array of potential consequences of anthropogenic climate change."

We now tie our results to the predicted trends in drought, as described in pages 1-2 of this document.

Line 32: And soil moisture? Drought? (for example, see references listed above)

Yes. We now clarify.

Lines 46-48: "The high temperatures and reduced soil moisture characteristic of drought play an important role in dust mobilization, since loss of vegetative cover during drought increases soil erosion (Archer and Predick, 2008; Bestelmeyer et al., 2018)."

Line 39: Here and throughout the paper it is unclear what the authors define as "climate change" and it is important to define it here. Do they just mean increased emissions? Or increased temperatures? Increased drought?

We have clarified our approach.

Lines 93-98: "In this study, we investigate the effects of climate change, increasing  $CO_2$  fertilization, and future land use practices on vegetation in southwestern North America, and we examine the response of dust mobilization due to these changes in vegetation. With regard to climate, we examine whether a shift to warmer, drier conditions by 2100 enhances dust mobilization in this region by reducing vegetation cover and exposing bare land."

Line 39-42: These sentences read like they should come towards the end of the Introduction.

Done.

Line 44-49: Where did these studies occur? It also seems that the estimates would depend largely on the particular region of study, since different regions may have different controlling factors.

These are all global studies.

Lines 70-72: "For example, Woodward et al., 2005 predicted a tripling of the global dust burden by 2100 relative to the present day, while other studies suggested a decrease in the global dust burden (e.g., Harrison et al., 2001, Mahowald and Luo, 2003 and Mahowald et al., 2006)."

Line 60: This study indicates the importance of drought, but again, it is not clear whether the impacts of increased drought is included in this study?

See response to next comment.

Line 69: If "climate-induced changes" includes the role of drought, it should be described here because it is unclear. If it is not, the authors need to address why this very important role was not considered.

As clarified on pages 1-2 of this document, our study takes into account the effect of changing temperatures and precipitation on vegetation, which in turn influences dust mobilization. Here is another place in the revised manuscript where we emphasize the role of drought.

Lines 153-157: "Nonetheless, the meteorological conditions predicted in the RCP8.5 scenario for 2100 align with previous studies projecting increased risk of drought in this region (e.g., Williams et al., 2020), and as we shall see, such conditions, in the absence of  $CO_2$  fertilization, result in decreased vegetation and greater dust mobilization.

Line 75: Like the abstract, the end of the Introduction would benefit from an implications statement, or some description of what these results could inform in terms of public policy or future studies.

We now add a description.

Lines 103-105: "Given the deleterious effects of airborne dust on human health, our dust projections under different climate scenarios have value for understanding the full array of potential consequences of anthropogenic climate change."

Line 77: An overall statement about the Methods section. It is difficult to follow, descriptions of some of the models and methods are scattered throughout the sections. The entire section would benefit from a streamlining and overall organization. It seems like the authors are giving a brief overview of the method at the beginning, which is fine, but in its present form it includes some details that leave the reader looking around for descriptions that aren't included until later. Perhaps leave the overview very general and then describe each step in more detail.

We have restructured the overview of the Methods section and now include a new subsection describing the GISS Model E (Section 2.1).

Line 85:86- Time periods aren't given, GISS is discussed here but then again in line 133 (maybe a separate "GISS" section, like the other models have?).

We now include information about the time period simulated by the GISS model.

Lines 132-133: "The simulations cover the years 1801 to 2100 at a spatial resolution of  $2^{\circ}$  latitude x 2.5° longitude."

Line 88: Again, what does "changes in climate" mean in this context?

We now clarify.

Lines 180-186. "For this study we follow Li et al., 2020, in linking meteorology from GISS-E2-R to LPJ-LMfire in order to capture the effects of climate change on vegetation. Meteorological fields from the GISS model include monthly mean surface temperature, diurnal temperature range, total monthly precipitation, number of days in the month with precipitation greater than 0.1 mm, monthly mean total cloud cover fraction, and monthly mean surface wind speed. Monthly mean lightning strike density, calculated using the GISS convective mass flux and the empirical parameterization of Magi, 2015, is also applied to LPJ-LMfire."

Lines 189-190. "LPJ-LMfire then simulates the response of natural vegetation to the 21st century trends in these meteorological fields and to increasing CO2."

**Line 93: How is "fine dust" defined (again, this comes later).**

As described above, we define fine dust in the Abstract and also now in the Introduction. Lines 35-37: "By causing respiratory and cardiovascular diseases, fine dust particles – i.e., those particles with diameter less than 2.5 microns – can have negative effects on human health..."

Line 100: We are in the Methods section?

Fixed.

Line 102-107: This seems misplaced, perhaps it should go in a separate "GISS" section?

We have a new section on the GISS model, with more detail on the simulation. For example, we now say the following.

Lines 133-135: "Changes in climate in the GISS model are driven by increasing greenhouse gases. In RCP4.5, CO2 concentrations increase to 550 ppm by 2100; in RCP8.5 the CO2 increases to 1960 ppm (Meinshausen et al., 2011)."

Line 109: What land use fields are included in the model and where do they come from? Some reference to this is included in line 237 but would be useful to know sooner. Where does the vegetation information come from? Is it representative of desert vegetation? Where does wildfire information come from and does it change over time? Do the meteorological anomalies characterize future drought?

We have revamped part of Section 2.2 on LPJ-LMfire.

Lines 180-191: "For this study we follow Li et al., 2020, in linking meteorology from GISS-E2-R to LPJ-LMfire in order to capture the effects of climate change on vegetation. Meteorological fields from the GISS model include monthly mean surface temperature, diurnal temperature range, total monthly precipitation, number of days in the month with precipitation greater than 0.1 mm, monthly mean total cloud cover fraction, and monthly mean surface wind speed. Monthly mean lightning strike density, calculated using the GISS convective mass flux and the empirical parameterization of Magi, 2015, is also applied to LPJ-LMfire. To downscale the  $2^{\circ} \times 2.5^{\circ}$  GISS meteorology to finer resolution for LPJ-LMfire, we calculate the 2010-2100 monthly anomalies relative to the average over the 1961-1990 period, and then add these anomalies to an observationally based climatology (Pfeiffer et al., 2013). Natural vegetation in LPJ-LMfire then simulates the response to the  $21^{\text{st}}$  century trends in these meteorological fields and to increasing CO2. We apply the same changes in CO2 concentrations as those applied to the GISS model."

Lines 192-196: "We overlay the changes in natural land cover with future land use scenarios from CMIP5 (LUH; Hurtt et al., 2011; http://tntcat.iiasa.ac.at/RcpDb/, last accessed on 17 July 2020). These scenarios include land used for crops, ranching (rangeland), and urban areas, all of which result in reduction in aboveground biomass, an increase in herbaceous relative to woody plants, and an increase in the extent of bare ground."

Lines 205-210: "The LPJ-LMfire simulations yield monthly timeseries of the leaf area indices (LAI) and fractional vegetation cover ( $\sigma_v$ ) for nine plant functional types (PFTs): tropical broadleaf evergreen, tropical broadleaf raingreen, temperate needleleaf evergreen, temperate broadleaf summergreen, boreal needleleaf evergreen, and boreal summergreen trees, as well as C3 and C4 grasses."

We convert the LAI from LPJ-LMfire to vegetation area index (VAI), and the result is generally comparable with satellite derived VAI for this region as well as observed land cover over the principle dust-producing regions.

Lines 30-37 (Supplement): "Figure S4 compares the differences in springtime VAI generated by LPJ-LMfire for the present day and that derived from 1-km reflectance data from the Advanced Very High Resolution Radiometer (AVHRR, Bonan et al., 2002). This satellite-based VAI is the default dataset in the DEAD module (Zender et al., 2003). The differences between these two VAI datasets are mostly small, within  $\pm 1$  m2 m-2, across southwestern North America, giving us confidence in the performance of LPJ-LMfire. In addition, we categorize the LPJ-LMfire simulated land cover types as trees and shrubs, grasses, and barren land (Figure S5). The high-dust emission region shown in Figure S3 is dominated by grass ecosystems and barren land, roughly consistent with observed land cover shown in the photos of four locations (southwest New Mexico, southeast New Mexico, west Texas, and northern Chihuahua state, Mexico) selected from the principle dust-producing regions in our study (Figure S5)."

Lines 42-45 (Supplement). "The dominant plant functional types in LPJ-LMfire in the southwestern North America include temperate needleleaf evergreen, temperate broadleaf summergreen, and C3 perennial grass, consistent with observed, present-day vegetation types (McClaran and van Devender, 1997)."

The LPJ-LMfire model simulates wildfire and its changes under future climate.

We now add more explanations.

Lines 172-179: "Wildfire in LPJ-LMfire depends on lightning ignition, and the simulation considers multiday burning, coalescence of fires, and the spread rates of different vegetation types. The effects of changing fire activity on vegetation cover are then taken into account (Pfeiffer et al., 2013; Sitch et al., 2003; Chaste et al., 2019). Li et al., 2020 predicted a ~50% increase in fire-season area burned by 2100 under scenarios of both moderate and intense future climate change over the western United States. However, the effects of changing fire on vegetation cover are insignificant in the grass and bare ground-dominated ecosystems of the desert Southwest, where the low biomass fuels cannot support extensive spread of fires."

As emphasized in pages 1-2 of this response document, the GISS meteorology in RCP8.5 by 2100 is indeed consistent with drought.

Lines 153-157: "Nonetheless, the meteorological conditions predicted in the RCP8.5 scenario for 2100 align with previous studies projecting increased risk of drought in this region (e.g., Williams

et al., 2020), and as we shall see, such conditions, in the absence of CO2 fertilization, result in decreased vegetation and greater dust mobilization."

Line 118: "Future land use scenarios applied follow CMIP5". Can the authors expand and define CMIP5? What all types of land use scenarios are included?

We now define CMIP5 and clarify what is meant by land use.

Lines 130-132: "..., configured for Phase 5 of the Coupled Model Intercomparison Project (CMIP5; https://esgf-node.llnl.gov/search/cmip5/, last accessed on 17 July 2020)."

Lines 192-196: "We overlay the changes in natural land cover with future land use scenarios from CMIP5 (LUH; Hurtt et al., 2011; http://tntcat.iiasa.ac.at/RcpDb/, last accessed on 17 July 2020). These scenarios include land used for crops, ranching (rangeland), and urban areas, all of which result in reduction in aboveground biomass, an increase in herbaceous relative to woody plants, and an increase in the extent of bare ground."

Line 121-122: Some discussion here regarding how the model accounts for hydrologic feedbacks, such as whether plants react to water limitation?

We have added more details about hydrologic feedbacks in the LPJ-LMfire model.

Lines 165-170: "More specifically, LPJ-LMfire simulates the impacts of photosynthesis, evapotranspiration, and soil water dynamics on vegetation structure and the population densities of different plants functional types (PFTs). The model considers the coupling of different ecosystem processes, such as the interactions between CO2 fertilization, evapotranspiration, and temperature, as well as the competition among different PFTs for water resources (e.g., precipitation, surface runoff, and drainage)."

Line 122: "...and analyze results over..." This sentence is redundant and unnecessary.

Deleted.

Line 125-128: Discussion of RCP4.5 and RCP4.8 seems out of order here.

We have moved the discussion of the RCPs to the beginning of Section 2.

Line 129-133: Redundant, see lines 85-87. Again, move the GISS information into a GISS section.

Done. We now have a new section on the GISS model, Section 2.1.

Line 161: How representative are these of desert plants in the Southwest?

Although cactuses are missing from LPJ-LMfire, overall, the simulated vegetation distribution and composition is consistent with observations. We now add more explanations.

Lines 241-244: "Of the nine PFTs, temperate needleleaf evergreen, temperate broadleaf evergreen, temperate broadleaf summergreen, and C3 grasses dominate the region, with temperate needleleaf evergreen having the highest LAI in spring. This mix of vegetation type is consistent with observations (e.g., McClaran and van Devender, 1997)."

Lines 42-45 (Supplement): "The dominant plant functional types in LPJ-LMfire in the southwestern North America include temperate needleleaf evergreen, temperate broadleaf summergreen, and  $C_3$  perennial grass, consistent with observed, present-day vegetation types (McClaran and van Devender, 1997)."

Line 165: I assume (based on equation 3) that 7 different PFTs are included to represent stem area index? What are they?

We consider the stem area index from just 7 PFTs.

Lines 235-237. "We also assume that  $C_3$  and  $C_4$  grasses have zero stem area to avoid overestimating VAI during the winter and early spring when such grasses are dead."

The term  $\sigma_v$  refers to fractional vegetation cover.

Lines 240-241: "...LAI is for the nine PFTs from LPJ-LMfire, but  $\sigma_v$  is for just seven PFTs, with  $\sigma_v$  for C3 and C4 grasses not considered."

Line 170: Are all plants represented here responsive to  $CO_2$  fertilization? How do the effects of drought, heat, and evapotranspiration offset gains in  $CO_2$  fertilization and can this be captured by the model? If not, it should be stated.

Yes, all PFTs in LPJ-LMfire respond to changing CO2. We show this through the series of sensitivity tests we performed (e.g., Figure S10).

Lines 170-172: "The different PFTs in LPJ-LMfire respond differently to changing  $CO_2$ , with  $CO_2$  enrichment preferentially stimulating photosynthesis in woody vegetation and  $C_3$  grasses compared to C4 grasses (Polley et al., 2013)."

We also have clarified the interactions considered by the model.

Lines 167-170: "The model considers the coupling of different ecosystem processes, such as the interactions between CO2 fertilization, evapotranspiration, and temperature, as well as the competition among PFTs for water resources (e.g., precipitation, surface runoff, and drainage)."

We have also made more clear in the Results section how CO2 fertilization may offset the impact of climate change.

Lines 291-293: "For the fixed-CO2 case, western New Mexico and northern Mexico show greater decreases in VAI, indicating how CO2 fertilization in the other two cases offsets the effects of the warmer and drier climate on vegetation in this region."

**Line 177: MERRA is mentioned here for the first time?**

We now clarify.

Lines 247-250: "We feed into the DEAD module both the VAI generated by LPJ-LMfire and meteorological fields from Modern-Era Retrospective analysis for Research and Applications (MERRA-2) at a spatial resolution of 0.5° latitude x 0.625° longitude (Gelaro et al., 2017)."

Line 202: Define "springtime"

Done in the Abstract and at the end of the Introduction.

Line 205: These boundaries are not shown on the figures and probably aren't important to mention here.

We have removed "National Forests and Parks."

Lines 283-285: "Strong enhancements (up to  $\sim 2.5 \text{ m}^2 \text{ m}^{-2}$ ) extend across much of Arizona, especially in the northwestern corner."

Line 237: This description of land use change would be helpful earlier.

Done.

Lines 62-67: "Land use practices, e.g., farming and ranching, industrial activities including mining, and urban sprawl, have changed dramatically over the southwestern North America in recent decades, with Arizona and New Mexico showing decreasing cropland area and northern Mexico experiencing increasing pasture area (Figure S1). Future land use practices could also influence the propensity for dust mobilization by disturbing crustal biomass (e.g., Belnap and Gillette, 1998)."

Lines 192-196: "We overlay the changes in natural land cover with future land use scenarios from CMIP5 (LUH; Hurtt et al., 2011; http://tntcat.iiasa.ac.at/RcpDb/, last accessed on 17 July 2020). These scenarios include land used for crops, ranching (rangeland), and urban areas, all of which result in reduction in aboveground biomass, an increase in herbaceous relative to woody plants, and an increase in the extent of bare ground."

Also, we now provide a figure showing changes in land use in the Supplement, and describe this in the text.

Lines 325-328: "Combined changes in land use are greater under RCP8.5 than RCP4.5, with large increases in RCP8.5 across Mexico but only modest changes in Arizona, New Mexico, and Texas (Figure S9). The increases in Mexico result in the fragmentation of forested landscapes and decrease VAI, especially in coastal forest regions and along the border with the United States."

Line 246: How is "desertification" defined? Does this imply anything about drought?

We have removed the reference to "desertification."

Line 257: How are "climate stresses" defined and quantified in the model? This implies impacts from drought and water stress on plants, but as mentioned before, this doesn't seem to be captured by the model? Should "temperature" be "temperate"?

Yes, climate stresses here do imply impacts from drought and water stress. We have fixed the typo. Lines 343-345: "These trends occur due to the climate stresses, e.g., warmer temperatures and decreased precipitation, that impair the growth of temperature broadleaf trees and C3 grasses. In this case, such stresses are not offset by CO2 fertilization (Figure S10)."

Line 264: What is the land use type shifting towards in these regions?

We have revised the sentence.

Lines 348-353: "Figure 3 also reveals that land use trends are a major driver of increased dust emissions along the ANM border and western Texas in RCP8.5, as crop- and rangelands expand in this region and temperature broadleaf trees decline (Hurtt et al., 2011). Similarly, the expansion of rangelands in northern Mexico in RCP8.5 reduces natural vegetation cover there (Hurtt et al., 2011), contributing to the increase of fine dust emissions by up to ~0.7 kg m-2 mon-1."

Line 277-278. I am not sure I understand this sentence. Land use is the driver, but climate change makes up the bulk of the increases?

We now clarify with several sentences.

Lines 361-363: "Results from GEOS-Chem in the fixed-CO2 case for RCP8.5 show that the concentrations of spring fine dust are significantly enhanced in the southeastern half of New Mexico and along the ANM border, with increases up to  $\sim 2.5 \,\mu g \, m^{-3}$  (Figure 4)."

Lines 364-369: "As Figure 3 implies, land use along the ANM border contributes to the increased dust emissions in that area, by up to ~ $0.7 \text{ kg m}^{-2} \text{ mon}^{-1}$ . Climate change impacts on natural vegetation, however, account for the bulk of the modeled increases in dust emissions in this scenario, by as much as ~ $1.2 \text{ kg m}^{-2} \text{ mon}^{-1}$  (Figure 2). The modeled wind fields, which are the same in all scenarios, transport the dust from source regions, leading to the enhanced concentrations across much of the domain, as seen in Figure 4."

Line 279: The authors seem to be implying that winds are also involved in these differences?

Yes, climate change leads to increased dust mobilization in the fixed CO2 RCP8.5 scenario, and the winds carry the dust across the region, as described in the response above.

Line 292: This wasn't specifically shown in the results (shifts in land surface type).

We have clarified this issue in the Results section. See Lines 346-353 quoted above.

Line 298-299: And this study doesn't include changes in wind speed, so it's hard to say that the differences between the Pu and Ginoux study are primarily due to the changes in vegetation.

We have clarified the comparison to Pu and Ginoux.

Lines 394-402: "In contrast, the statistical model of Pu and Ginoux, 2017 estimated a 2% decrease in the springtime frequency of extreme dust events in the Southwest U.S., driven mainly by reductions in bare ground fraction and wind speed. Like Pu and Ginoux, 2017, we also find that dust emissions decrease across a broad region of the Southwest when CO2 fertilization is taken into account, as shown in Figure 2. Pu and Ginoux, 2017 relied on limited data for capturing the sensitivity of dust event frequency to land cover in this region, and neither that study nor Achakulwisut et al., 2018 considered changes in land use, as we do here. The role of changing wind speed, however, is not included in our study, but could be tested in future work."

Line 308: So that I am understanding what is presented in the Table, CO2 fertilization would correspond to "fixed land use" but I don't see 30% or 64% in the table?

We now clarify this statement.

Lines 407-410: "Correspondingly, in the RCP4.5 scenario for 2100, CO2 fertilization enhances VAI by 30% in the all-factor case compared to the fixed-CO2 case (1.07 m2m-2 vs. 0.79 m2m-2); in RCP 8.5, the 2100 enhancement is 64% (1.11 m2m-2 vs. 0.55 m2m-2), as shown in Table 1."

Line 312-213: But, as stated previously, it is unclear whether future drought is accounted for, or whether the role of increased temperature and water stress on whether plants are responsive to  $CO_2$  fertilization is addressed. This seems like an important question the authors need to address, as it could change the directions of trends in dust emission. The authors need to discuss how or whether this was accounted for.

We have clarified the role of meteorological variables, including drought, as described on pages 1-2 of this document. We also now make clear that the coupling between CO2, water stress, and temperature is considered. New text is shown on Lines 160-170 (described above).

Line 367: References: There appears to be formatting inconsistencies with several of the references. I encourage the authors to check their reference manager settings (e.g., line 396, 399, 417, 433, 435, etc.). In addition, "doi's" were not included for any of the references.

We have updated the references and added DOIs for some of the references. Final corrections will be competed in the proofreading phase.

Line 486: Figure 1: This is the first time land use is referred to as "anthropogenic" and would benefit from a description of what this means (in text).

Land use is by definition anthropogenic. We acknowledge that the term "anthropogenic land use" is redundant and have fixed it in multiple places in the manuscript. We now describe land use in more detail (Lines 62-65), as mentioned above in this document.

Line 517: In the "a" description, include whether "2010" is the first year in the 5 year slice.

We now clarify this detail.

Lines 725-726: "Each time slice represents 5 years (i.e., 2011-2015 represents the 2010 time slice and 2095-2099 represents the 2100 time slice)."

Archer and Predick, 2008, "Climate change and ecosystems of the Southwestern United States", Rangelands, 30(3):23-28

Cited.

MacDonald, G.M., 2010 "Water, climate change, and sustainability in the Southwest", PNAS, 107(50).

**Cited.**

Prein et al., 2016, "Running dry: The U.S. Southwest's drift into a drier climate state", GRL, 43, doi:10.1002/2015GL066727.

**Cited.**

Stahle, D.W. 2020, "Anthropogenic megadrought", Science, 368 (6488).

Cited.

Williams, A. P., et al., 2020, "Large contribution from anthropogenic warming to an emerging North American megadrought", Science, 368 (314-318).

Cited.

**Author Response to Reviewer #2**

The authors present a study of how dust emissions across southwestern US states could respond to projected climate changes, elevated atmospheric CO2 and land use change. Projected climate changes are assessed for two Representative Concentration Pathways (RCP 4.5 and 8.5) representing moderate and continued increases in greenhouse gas concentrations through the 21st century. The effects of the climate projections on surface erodibility are represented through a dynamic vegetation model that is linked to a dust emission scheme and the GEOS-Chem chemical transport model. The general subject matter of the manuscript and approach taken is consistent with regional dust modelling approaches today. Linking a dynamic vegetation model to a dust model to investigate projected climate changes is novel, not straightforward, and has potential to

provide new insights into the effects (and interactions) of dust emission under changing land uses and climate.

Overall, my assessment is that, while the subject matter is timely, the manuscript has a number of shortcomings that reduce the relevance of the work and confidence that the conclusions are adequately supported by the approach. These include:

1) While the first paragraph of the Introduction seeks to establish the relevance of the study, this is done only at a very high level and specific research and management impetus are not provided. This high-level treatment of the rationale for the work is carried throughout the manuscript, with the text rarely going deeper than general drivers and responses to justify why the work is important, how it can have impact, who it may have impact for, or how any of the processes and interactions between vegetation, land use and climate actually work and may influence future dust emissions. The superficial treatment of these important elements reduces the impact of the work. Adding detail to these elements would give the work more weight and enable the authors to show exactly what the new insights are that they provide, how they are relevant, and where key uncertainties are.

We thank the reviewer for these thoughtful comments, which we break down into the components below.

**1a. Why is the work important, and how can it have impact?**

Lines 10-13: "Climate models predict a shift toward warmer and drier environments in southwestern North America over the 21st century. The consequences of climate change for dust mobilization and concentrations are unknown, but could have large implications for human health, given connections between dust inhalation and disease."

Lines 96-98: "With regard to climate, we examine whether a shift to warmer, drier conditions by 2100 enhances dust mobilization in this region by reducing vegetation cover and exposing bare land."

Lines 103-105: "Given the deleterious impacts of airborne dust on human health, our dust projections under different climate scenarios have value in understanding the full array of potential consequences of anthropogenic climate change."

**1b. Whom does the work has impact on?**

Lines 29-31: "Our results have implications for human health, especially for the health of the indigenous people who make up a large percentage of the population in this region."

Lines 457-462: "In the absence of increased CO2 fertilization, our work suggests that vegetated cover will contract in response to the warmer, drier climate, exposing bared ground and significantly increasing dust concentrations by 2100. In this way, dust enhancement could impose a potentially large climate penalty on PM2.5 air quality, with consequences for human health across much of southwestern North America."

Lines 463-468: "Our finding of the potential for an increased dust burden in the future atmosphere has special relevance for environmental justice in this region, where much of the current population is of Native American and/ or Latino descent. For example, in New Mexico, 10% of the population is Native American and 50% identifies as either Hispanic or Latino. By some measures, New Mexico has also one of highest poverty rates of the United States (https://www.census.gov/quickfacts/NM, last accessed on August 20, 2020)."

**1c. How do the processes and interactions between vegetation, land use, and climate actually work and how do they influence dust mobilization?**

Lines 96-98: "With regard to climate, we examine whether a shift to warmer, drier conditions by 2100 enhances dust mobilization in this region by reducing vegetation cover and exposing bare land."

Lines 165-179: "More specifically, LPJ-LMfire simulates the impacts of photosynthesis, evapotranspiration, and soil water dynamics on vegetation structure and the population densities of different plants functional types (PFTs). The model considers the coupling of different ecosystem processes, such as the interactions between CO2 fertilization, evapotranspiration, and temperature as well as the competition among different PFTs for water resources (e.g., precipitation, surface runoff, and drainage). The different PFTs in LPJ-LMfire respond differently to changing CO2, with CO2 enrichment preferentially stimulating photosynthesis in woody vegetation and C3 grass compared to C4, (Polley et al., 2013). Wildfire in LPJ-LMfire depends on lightning ignition, and the simulation considers multiday burning, coalescence of fires, and the spread rates of different vegetation types. The effects of changing fire activity on vegetation cover are then taken into account (Pfeiffer et al., 2013; Sitch et al., 2003; Chaste et al., 2019). Li et al., 2020 predicted a ~50% increase in fire-season area burned by 2100 under scenarios of both moderate and intense future climate change over the western United States. However, the effects of changing fire on vegetation cover are insignificant in the grass and bare ground-dominated ecosystems of the desert Southwest, where the low biomass fuels cannot support extensive spread of fires."

Lines 414-423: "In summary, we find that as atmospheric CO2 levels rise vegetation growth is enhanced and dust mobilization decreases, offsetting the impacts of warmer temperatures and reduced rainfall, at least in some areas. These results are consistent with evidence that CO2 fertilization is already occurring in arid or semiarid environments like southwestern North America (Donohue et al., 2013; Haverd et al., 2020). In such environments, water availability is the dominant constraint on vegetation growth, and the recent increases in atmospheric CO2 may have reduced stomatal conductance and limited evaporative water loss. The effects of CO2 fertilization on vegetation growth are uncertain, however, and may be attenuated by the limited supply of nitrogen and phosphorus in soil (Wieder et al., 2015). These nutritional constraints vary greatly among different PFTs (Shaw et al., 2002; Nadelhoffer et al., 1999)."

Lines 457-460: "In the absence of increased CO2 fertilization, our work suggests that vegetated cover will contract in response to the warmer, drier climate, exposing bared ground and significantly increasing dust concentrations by 2100."

2) A focus of the manuscript is establishing how future vegetation and land use changes may influence dust emissions. However, the authors have not grounded the manuscript in the present situation – What types of vegetation communities are there across the study area? What types of land use changes are occurring today? How important is land use versus land management? How do these present changes relate to the modeled vegetation and land use change scenarios? How are the vegetation communities changing today? What are the implications of vegetation change trajectories today for future responses to elevated CO2, climate change, and land use? How are these changes related to and influence aeolian processes? By not addressing these questions, the work presents as a typical dust modelling study and/but detached from reality. Expanding the Introduction and Discussion sections is needed to ground the work 'in the real world' and could help the authors demonstrate the relevance and contribution of the study (point #1 above).

Again, we break down the reviewer's questions into components.

**2a. What types of vegetation communities are there across the study area?**

Lines 49-50: "Southwestern North America is covered by desert grassland, perennial grassland, savanna, desert scrub, and grassy shrublands or woodlands (McClaran and Van Devender, 1997)."

Lines 241-244: "Of the nine PFTs, temperate needleleaf evergreen, temperate broadleaf evergreen, temperate broadleaf summergreen, and C3 grasses dominate the region, with temperate needleleaf evergreen having the highest LAI in spring. This mix of vegetation type is consistent with observations (e.g., McClaran and van Devender, 1997)."

Figure S4 compares the differences between springtime VAI simulated by LPJ-LMfire and that derived from 1-km satellite data in southwestern North America. Figure S5 further compares LPJ simulated vegetation types with observed land cover for four selected locations across the principle dust-producing regions.

Lines 30-41 (Supplement): "Figure S4 compares the differences in springtime VAI generated by LPJ-LMfire for the present day and that derived from 1-km reflectance data from the Advanced Very High Resolution Radiometer (AVHRR, Bonan et al., 2002). This satellite-based VAI is the default dataset in the DEAD module (Zender et al., 2003). The differences between these two VAI datasets are mostly small, within  $\pm 1$  m2 m-2, across southwestern North America, giving us confidence in the performance of LPJ-LMfire. In addition, we categorize the LPJ-LMfire simulated land cover types as trees and shrubs, grasses, and barren land (Figure S5). The high-dust emission region shown in Figure S3 is dominated by grass ecosystems and barren land, roughly consistent with observed land cover shown in the photos of four locations (southwest New Mexico, southeast New Mexico, west Texas, and northern Chihuahua state, Mexico) selected from the principle dust-producing regions in our study (Figure S5)."

Lines 42-45 (Supplement): "The dominant plant functional types in LPJ-LMfire in the southwestern North America include temperate needleleaf evergreen, temperate broadleaf summergreen, and C3 perennial grass, roughly consistent with observed, present-day vegetation types (McClaran and van Devender, 1997)."

**2b. What types of land use changes are occurring today?**

Lines 62-65: "Land use practices, e.g., farming and ranching, industrial activities including mining, and urban sprawl, have changed dramatically over the southwestern North America in recent decades, with Arizona and New Mexico showing decreasing cropland area and northern Mexico experiencing increasing pasture area (Figure S1)."

**2c. How important is land use versus land management?**

In our study, land use refers to the human use of land -e.g., establishing and maintaining croplands or settlements. Land management typically refers to how humans manage the land once natural vegetation has been altered -e.g., through fertilizer use, crop rotation, agricultural fires, or fire suppression. In our simulations, fire is not allowed to occur on cropland and rangeland, so we do have some land management. On the other hand, we do not account for stocking densities on rangeland, which when mismanaged, can lead to reduction of vegetation cover and enhanced dust emissions.

Lines 192-201. "We overlay the changes in natural land cover with future land use scenarios from CMIP5 (LUH; Hurtt et al., 2011; http://tntcat.iiasa.ac.at/RcpDb/, last accessed on 17 July 2020). These scenarios include land used for crops, ranching (rangeland), and urban areas, all of which result in reduction in aboveground biomass, an increase in herbaceous relative to woody plants, and an increase in the extent of bare ground. The present-day land use in the LUH dataset is taken from the HYDE database v3.1 (Goldewijk, 2001; Goldewijk et al., 2010), which in turn is based on array of sources, including satellite observations and government statistics. In RCP8.5, the extent of crop- and rangeland cover increases by ~30% in Mexico but decreases by 10-20% over areas along Mexico's northern border in the U.S. (Hurtt et al., 2011). Only minor changes in land use practices by 2100 are predicted under RCP4.5 (Hurtt et al., 2011)."

**2d. How do present changes in land use relate to the modeled vegetation and land use change scenarios?**

We validate the present-day land cover in LPJ-LMfire, as described in #2a above, and we discuss the extent of present-day land use and recent changes in #2b above. The source of present-day land use is the HYDE database v3.1.

Lines 196-198: "The present-day land use in the LUH dataset is taken from the HYDE database v3.1 (Goldewijk, 2001; Goldewijk et al., 2010), which in turn is based on array of sources, including satellite observations and government statistics."

**2e. How are the vegetation communities changing today? What are the implications of vegetation change trajectories today for future responses to elevated CO2, climate change, and land use?**

We now comment on recently observed changes in land cover in response to drought.

Lines 384-390: "Our findings of decreasing VAI with future climate change are consistent with observed trends in vegetation during recent droughts in this region. For example, Breshears et al., 2005 documented large-scale die-off of overstory trees across southwestern North America in 2002-2003 in response to short-term drought accompanied by bark beetle infestations. Similarly, during a multi-year (2004-2014) drought in southern Arizona, Bodner and Robles, 2017 found that the spatial extent of both C4 grass cover and shrub cover decreased in the southeastern part of that state."

3) The modeled vegetation changes appear unconnected to vegetation changes occurring across southwestern US landscapes today and are not adequately represented in the dust model. As described in Sections 2.2 and 2.3, the DEAD model is used to estimate dust emissions, with vegetation effects represented through a linear adjustment term Av that is calculated from VAI that is the sum of leaf and stem area indices. This approach makes two assumptions that are inconsistent with the physics of aeolian transport and drag partition theory: 1) fractional vegetation cover adequately represents lateral surface aerodynamic sheltering – ergo structural changes in surface roughness due to changing vegetation were not represented while they are likely to have a greater influence on dust emissions than fractional ground cover (Av), and 2) adjustments to the fractional vegetation cover can be made through a dynamic vegetation model (to represent vegetation change) that are separate to the dust model drag partition scheme and its use of aerodynamic roughness lengths (z0) - creating a functional disparity in how vegetation is represented in different parts of the model. I identify these issues in full recognition of the difficulty of accurately representing future vegetation change in a dust model. However, these two assumptions also potentially undermine the validity of the model experiments and so need to be addressed transparently. Further, what are the implications of the model parameterization for the rigor of the results? How much confidence can we have in the outcomes of the study? Where are the gaps that need to be addressed? Turning this challenge into a positive – what insights does this work provide for how future research can address interactions among climate change, vegetation change, land use and dust emissions?

Again, we address the comments by component.

**3a.** The reviewer states that the modeled vegetation changes appear "unconnected" to observed vegetation changes occurring across southwestern US landscapes today.**

We validate the present-day land cover, as described in #2a above. Present-day land-use is from the HYDE database, which in turn depends on satellite observations and government statistics, as described in #2d above.

**3b. The reviewer points out that fractional vegetation cover may not adequately represent lateral surface aerodynamic sheltering. This is a common weakness among dust models, and we now acknowledge this shortcoming in the Discussion.**

Lines 430-440. "Other uncertainties in our study can be traced to the dust simulation. The different vegetation types in our model are quantified as fractions of gridcells, which have relatively large spatial dimensions of ~50 km × 60 km. This means the model cannot capture the spatial heterogeneity of land cover, and the aerodynamic sheltering effects of vegetation on wind erosion

are neglected, as they are in most 3-D global model studies. Such sheltering could play a large role in dust mobilization (e.g., Liu, 1990). New methods involving satellite observations of surface albedo promise to improve understanding of the effects of aerodynamic sheltering on dust mobilization, at least for the present-day (Chappell and Webb, 2016; Webb and Pierre, 2018). Implementation of aerodynamic sheltering in simulations of future climate regimes would need to account for fine-scale spatial distributions of vegetation."

**3c.** Finally, the reviewer points out a "functional disparity" in our approach, with vegetation changes applied to the calculation of VAI but not to that of aerodynamic roughness length. We now acknowledge this disconnect.**

Lines 253-256: "The scheme assumes that the vertical flux of dust is proportional to the horizontal saltation flux, which in turn depends on surface friction velocity and the aerodynamic roughness length  $Z_0$ . As recommended by Zender et al., 2003, and consistent with Fairlie et al. (2007) and Ridley et al. (2013), we uniformly set  $Z_0$  to 100 µm across all dust candidate grid cells."

Lines 440-443: "In addition, as recommended by Zender et al. (2003), we apply a globally uniform surface roughness  $Z_0$  in the model, which means that the impact of changing vegetation conditions on friction velocity is not taken into account. Future work could address this weakness by varying friction velocity according to vegetation type."

While we have not explored the entire range of parameter uncertainty in the model, we do tackle the principle drivers of vegetation/dust change by running sensitivity tests with fixed climate and CO2 and land use. These scenarios allow us to show the range of potential possible outcomes.

4) Literature cited is constrained to dust modelling studies and a few supporting studies related to the vegetation and climate modelling. In addressing my concerns above, the authors could draw on the rich and diverse literature addressing vegetation and land use changes, and their interactions with aeolian processes, across the southwestern US.

We have added a lot of citations that address vegetation and land use change. Here are some examples:

- Andreadis, K. M., E. A. Clark, A. W. Wood, A. F. Hamlet, and D. P. Lettenmaier (2005), Twentieth-century drought in the conterminous United States, J. Hydrometeorology, 6(6), 985–1001.
- Belnap, J., and D. A. Gillette (1998), Vulnerability of desert biological soil crusts to wind erosion: the influences of crust development, soil texture, and disturbance, *Journal of Arid Environments*, *39*, 133–142.
- Bodner, G. S., and M. D. Robles (2017), Enduring a decade of drought: Patterns and drivers of vegetation change in a semi-arid grassland, *Journal of Arid Environments*, *136*(C), 1–14, doi:10.1016/j.jaridenv.2016.09.002.
- Breshears, D. D. et al. (2005), Regional vegetation die-off in response to global-change-type drought, *Proc. Natl. Acad. Sci.*, 102(42), 15144–15148, doi:10.1073/pnas.0505734102.
- Chappell, A., and N.P. Webb (2016), Using albedo to reform wind erosion modelling, mapping and monitoring, Aeolian Research, 23, 63-78, doi:10.1016/j.aeolia.2016.09.006

- Chaste, E., M. P. Girardin, J. O. Kaplan, Y. Bergeron, and C. Hély (2019), Increases in heatinduced tree mortality could drive reductions of biomass resources in Canada's managed boreal forest, *Landscape Ecology*, *34*(2), 403–426, doi:10.1007/s10980-019-00780-4.
- Donohue, R. J., M. L. Roderick, T. R. McVicar, and G. D. Farquhar (2013), Impact of CO2 fertilization on maximum foliage cover across the globe's warm, arid environments, *Geophysical Research letters.*, 40(12), 3031–3035, doi:10.1002/grl.50563.
- Edwards, B. L., N. P. Webb, D. P. Brown, E. Elias, D. E. Peck, F. B. Pierson, C. J. Williams, and J. E. Herrick (2019), Climate change impacts on wind and water erosion on US rangelands, *Journal of Soil and Water Conservation*, 74(4), 405–418, doi:10.2489/jswc.74.4.405.
- Fairlie, T. D., D. J. Jacob, and R. J. Park (2007), The impact of transpacific transport of mineral dust in the United States, *Atmos. Env.*, 41(6), 1251–1266, doi:10.1016/j.atmosenv.2006.09.048.
- Haverd, V., B. Smith, J. G. Canadell, M. Cuntz, S. Mikaloff Fletcher, G. Farquhar, W. Woodgate, P. R. Briggs, and C. M. Trudinger (2020), Higher than expected CO2 fertilization inferred from leaf to global observations, *Global Change Biology*, 26(4), 2390–2402, doi:10.1111/gcb.14950.
- Klein Goldewijk, K. (2001), Estimating global land use change over the past 300 years: The HYDE Database, *Global Biogeochem. Cycles*, *15*(2), 417–433.
- Klein Goldewijk, K., A. Beusen, G. Van Drecht, and M. De Vos (2011), The HYDE 3.1 spatially explicit database of human-induced global land-use change over the past 12,000 years, *Global Ecology and Biogeography*, 20(1), 73–86, doi:10.1111/j.1466-8238.2010.00587.x.
- Liu, S. J., H. I. Wu, R. L. Lytton, and P. J. Sharpe (1990), Aerodynamic sheltering effects of vegetative arrays on wind erosion: A numerical approach, *Journal of Environmental Management*, 30(3), 281–294.
- Van Loon, A. F. et al. (2016), Drought in a human-modified world: Reframing drought definitions, understanding, and analysis approaches, *Hydrol. Earth Syst. Sci.*, 20(9), 3631–3650, doi:10.5194/hess-20-3631-2016.
- Webb, N. P., and C. Pierre (2018), Quantifying anthropogenic dust emissions, *Earth's Future*, 6(2), 286–295, doi:10.1002/2017EF000766.
- Williams, A. P. et al. (2013), Temperature as a potent driver of regional forest drought stress and tree mortality, *Nature Climate Change*, *3*, 292–297, doi:10.1038/nclimate1693.

Some specific concerns are as follows:

Line 65: Given the focus of the manuscript on land use and vegetation change as a driver of changing dust emissions, the introduction would benefit from inclusion of a review paragraph/synthesis of the types of vegetation and the trajectories of these ecosystems across the southwest today. This is likely to have important implications for trends in dustiness, with pervasive vegetation changes influencing surface aerodynamics and wind erosivity. The authors might also comment on the likely sensitivity of these vegetation communities to elevated  $CO_2$ . See for example references within:

Bestelmeyer et al., 2018. The Grassland-Shrubland Regime Shift in the Southwestern United States: Misconceptions and Their Implications for Management. Bioscience 68, 678-690.

Edwards et al., 2019. Climate change impacts on wind and water erosion on US rangelands. Journal of Soil and Water Conservation. Vol. 74, 405-418. doi:10.2489/jswc.74.4.405.

We have revised the introduction and now cite these recommended papers.

Lines 49-55: "Southwestern North America is covered by desert grassland, perennial grassland, savanna, desert scrub, and grassy shrublands or woodlands (McClaran and Van Devender, 1997). In recent decades, a gradual transition from grasslands to shrubland has been observed across much of this region, with increased aridity, atmospheric CO2 enrichment, and livestock grazing all possibly playing a role in this trend (Bestelmeyer et al., 2018). Future climate change may further prolong this transition, especially since shrubs fare better than grasses under a climate regime characterized by large fluctuations in annual precipitation (Bestelmeyer et al., 2018; Edwards et al., 2019)."

Lines 311-314. "As predicted by previous studies (Bestelmeyer et al., 2018; Edwards et al., 2019),  $C_3$  perennial grasses ( $C_3$ gr) in this case decrease across a large swath extending from Arizona through Mexico, showing the impacts of warmer temperatures and reduced precipitation, as well as (for Mexico) land use change."

We have added a discussion on the sensitivity of vegetation to elevated CO2 as:

Lines 59-62: "On the other hand, elevated CO2 concentrations in the future atmosphere could increase photosynthesis and decrease transpiration of some vegetation species, allowing for more efficient water use and enhancing growth (Poorter and Perez-Soba, 2002; Polley et al., 2013)."

Lines 170-172: "The different PFTs in LPJ-LMfire respond differently to changing CO2, with CO2 enrichment preferentially stimulating photosynthesis in woody vegetation and C3 grasses compared to C4 grasses (Polley et al., 2013)."

Line 110: How important is fire in the study area, if at all for the changes under investigation? Supporting references would help.

The LPJ-LMfire model considers the impact of wildfire on vegetation, which could be significant under a warmer and drier climate.

We now add more explanations.

Lines 141-145: "In addition, lightning strike densities decrease by ~0.006 strikes km-2 d-1 over Arizona in RCP4.5, but increase by the same magnitude in this region in RCP8.5 (Li et al., 2020). Lightning strikes play a major role for wildfire ignition in this region, while wildfires may influence landscape succession (e.g., Bodner and Robles, 2017)."

Lines 172-179: "Wildfire in LPJ-LMfire depends on lightning ignition, and the simulation considers multiday burning, coalescence of fires, and the spread rates of different vegetation types. The effects of changing fire activity on vegetation cover are then taken into account (Pfeiffer et al., 2013; Sitch et al., 2003; Chaste et al., 2019). Li et al., 2020 predicted a ~50% increase in fire-season area burned by 2100 under scenarios of both moderate and intense future climate change

over the western United States. However, the effects of changing fire on vegetation cover are insignificant in the grass and bare ground-dominated ecosystems of the desert Southwest, where the low biomass fuels cannot support extensive spread of fires."

Lines 314-316: "Increased fire activity also likely plays a role in the simulated decreases of forest cover and  $C_3$  grasses for RCP8.5 in southern Arizona, where fires together with drought may have affected landscape succession (Williams et al., 2013; Bodner and Robles, 2017)."

Line 125: It would be helpful if the authors can define what they mean by vegetation structure. Is this purely geometric (e.g., height, width of plants), or does this include spatial patterns in landscapes?

We now clarify.

Lines 163-164: "Here 'vegetation structure' refers to vegetation types and the spatial patterns in landscapes."

Line 157: The authors use an estimate of fractional vegetation cover to linearly account for vegetation effects which are predominantly lateral and non-linear for saltation flux and dust emission. While working within the constraints of the DEAD model, the authors should recognize the limitations of this approach and implications for the sensitivity of the model to vegetation change and accuracy of its representation of dust emission responses.

We now clarify this limitation, as also described above.

Lines 253-256: "The scheme assumes that the vertical flux of dust is proportional to the horizontal saltation flux, which in turn depends on surface friction velocity and the aerodynamic roughness length  $Z_0$ . As recommended by Zender et al., 2003, and consistent with Fairlie et al., 2007 and Ridley et al., 2013, we uniformly set  $Z_0$  to 100 µm across all dust candidate grid cells."

Lines 440-443: "In addition, as recommended by Zender et al., 2003, we apply a globally uniform surface roughness  $Z_0$  in the model, which means that the impact of changing vegetation conditions on friction velocity is not taken into account. Future work could address this weakness by varying friction velocity according to vegetation type."

Line 161: How representative are these classes of vegetation communities across the southwest? How do they relate to actual patterns of vegetation? For reference, the authors might look at NRCS ecological site descriptions across the study area.

As mentioned above, we now better describe present-day vegetation in this region.

Lines 49-50: "Southwestern North America is covered by desert grassland, perennial grassland, savanna, desert scrub, and grassy shrublands or woodlands (McClaran and Van Devender, 1997)."

Lines 196-198: "The present-day land use in the LUH dataset is taken from the HYDE database v3.1 (Goldewijk, 2001; Goldewijk et al., 2010), which in turn is based on array of sources, including satellite observations and government statistics."

Lines 241-244: "Of these nine PFTs, temperate needleleaf evergreen, temperate broadleaf evergreen, temperate broadleaf summergreen, and C3 grasses dominate the region, with temperate needleleaf evergreen having the highest LAI in spring. This mix of vegetation type is consistent with observations (e.g., McClaran and van Devender, 1997)."

Please also see the validation of the modeled VAI as described above and in the Supplement (Lines 26-41) and Figures S4 and S5.

Line 166: Although, during the first half of spring in the desert southwest,  $C_3$  shrubs (e.g., Prosopis glandulosa) may not have leaves such that the main aerodynamic effect is provided by branches and stems. It would be instructive to link actual plant phenology in the study area to what is/is not represented in the vegetation model.

What the reviewer requests would be challenging to carry out in this model study, but we do now acknowledge this shortcoming.

Lines 45-48 (Supplement): "We acknowledge, however, that with only nine PFTs, LPJ-LMfire cannot capture the phenology of all plant species, which could in turn introduce error into our dust calculations. Still, the relatively good match of modeled springtime VAI with that observed is encouraging."

Line 174: How did the authors parameterize the drag partition scheme and represent land use change effects in the dust model? In DEAD, these are represented through the MB95 drag partition scheme, with aerodynamic roughness lengths (z0) assigned to land cover classes. As dust emission is a lateral process, z0 and the drag partition should have a larger effect on dust emission than fractional cover via VAI. If z0 was not changed consistently with the fractional cover of vegetation, the model would represent an inconsistent vegetation effect and would likely not capture the nature of dust emission responses to the examined scenarios.

As mentioned on the previous page, we apply a uniform aerodynamic roughness length  $Z_0$ , and we acknowledge this limitation in the Discussion. See lines 250-253 and lines 440-443 in the revised main text.

Line 180: Do the authors mean saltation, or dust emission? Although a general term, dust shouldn't be saltating.

Fixed.

Line 250-253: "Following Ridley et al., 2013, we characterize subgrid-scale surface winds as a Weibull probability distribution, which allows saltation even when the grid-scale wind conditions are below some specified threshold speed."

Line 192: Can the authors describe the implications of not changing wind speed? Would you anticipate wind speed changes in response to regional vegetation (roughness) change and changes in synoptic meteorology?

We now clarify the implications of not considering changing winds in the future simulation.

Lines 145-148: "Finally, future surface wind speeds do not change significantly under RCP4.5, but increase slightly by ~4% across southwestern North America under RCP8.5 by 2100 (not shown). The increasing winds in RCP8.5 will influence the spread of fires in our study, but will not affect the simulated dust fluxes directly, as described in more detail below."

Lines 270-272: "In other words, we neglect the direct effects of future changes in wind speeds on dust mobilization, allowing us to focus instead on the indirect effects of changing vegetation on dust."

Lines 443-447: "Finally, our study focuses only on the effect of changing vegetation on dust mobilization and does not take into account how changing wind speeds or drier soils in the future atmosphere may more directly influence dust. Given the slight increase in monthly mean winds in RCP8.5 by 2100, future dust emissions in this scenario could be underestimated."

Line 201: Discussion point - what about changes in seasonality due to changes in plant phenological changes due to species change and change in the timing of warming and precipitation? This is partially addressed in the results, but would benefit from further discussion linked to actual plant communities.

Lines 316-319: "We also investigate trends in LAI for different months in spring from the present day to 2100. We find that the greatest percentage decreases in TeBS and  $C_3$  grasses occur in May, consistent with the largest decreases in precipitation in that month (not shown)."

Line 235: The effect of vegetation on dust emission shouldn't be reduced to growth as it is the kinds and proportions of vegetation in the landscape that influence surface aerodynamic roughness and spatial patterns of dust emission. These changes aren't represented in the model, but do need to be addressed by the authors.

As mentioned above, we now amended the text.

Lines 430-440. "Other uncertainties in our study can be traced to the dust simulation. The different vegetation types in our model are quantified as fractions of gridcells, which have relatively large spatial dimensions of ~50 km  $\times$  60 km. This means the model cannot capture the spatial heterogeneity of land cover, and the aerodynamic sheltering effects of vegetation on wind erosion are neglected, as they are in most 3-D global model studies. Such sheltering could play a large role in dust mobilization (e.g., Liu et al., 1990). New methods involving satellite observations of surface albedo promise to improve understanding of the effects of aerodynamic sheltering on dust mobilization, at least for the present-day (Chappell and Webb, 2016; Webb and Pierre, 2018). Implementation of aerodynamic sheltering in simulations of future climate regimes would need to account for fine-scale spatial distributions of vegetation."

Line 246: Can the authors define what they mean by desertification, and how this differs to the vegetation changes (grass-shrub transitions) that have already occurred over much of this region? e.g., for reference see Bestelmeyer, B.T., Okin, G.S., Duniway, M.C., Archer, S.R., Sayre, N.F., Williamson, J.C., Herrick, J.E., 2015. Desertification, land use, and the transformation of global drylands. Frontiers in Ecology and the Environment 13, 28-36.

We have removed this reference to desertification.

Line 269: What conditions would make  $CO_2$  of limited importance? Can the authors explain and expand on this in the Discussion? Will  $CO_2$  be the main driver of vegetation change, or are other factors likely to be more important/have been important in the past that are likely to influence future trends? (e.g., vegetation state transitions driven in part by land management, not just land use)

First, in the Introduction, as discussed above, we have described in more detail the main factors driving dust concentrations.

In the Discussion, we now clarify the uncertainties in the effects of CO2 fertilization.

Lines 414-423: "In summary, we find that as atmospheric  $CO_2$  levels rise vegetation growth is enhanced and dust mobilization decreases, offsetting the impacts of warmer temperatures and reduced rainfall, at least in some areas. These results are consistent with evidence that  $CO_2$ fertilization is already occurring in arid or semiarid environments like southwestern North America (Donohue et al., 2013; Haverd et al., 2020). In such environments, water availability is the dominant constraint on vegetation growth, and the recent increases in atmospheric  $CO_2$  may have reduced stomatal conductance and limited evaporative water loss. The effects of  $CO_2$  fertilization on vegetation growth are uncertain, however, and may be attenuated by the limited supply of nitrogen and phosphorus in soil (Wieder et al., 2015). These nutritional constraints vary greatly among different PFTs (Shaw et al., 2002; Nadelhoffer et al., 1999)."

Lines 453-462: "Given the many uncertainties, it is challenging to gauge which of the three factors investigated here – climate impacts on vegetation, CO2 fertilization, or land use change – will play the dominant role in driving future changes in dust emissions and concentrations. This study thus brackets a range of possible dust scenarios for the southwestern North America, with the simulation without CO2 fertilization placing an upper bound on dust emissions. In the absence of increased CO2 fertilization, our work suggests that vegetated cover will contract in response to the warmer, drier climate, exposing bared ground and significantly increasing dust concentrations by 2100. In this way, dust enhancement could impose a potentially large climate penalty on PM2.5 air quality, with consequences for human health across much of southwestern North America."

Line 278: It would help for the authors to expand on this point about wind as my understanding is that wind speeds were not adjusted for climate changes in the scenarios/ simulations.

We now clarify.

Lines 364-369: "As Figure 3 implies, land use along the ANM border contributes to the increased dust emissions in that area, by up to ~ $0.7 \text{ kg m}^{-2} \text{ mon}^{-1}$ . Climate change impacts on natural vegetation, however, account for the bulk of the modeled increases in dust emissions in this scenario, by as much as ~ $1.2 \text{ kg m}^{-2} \text{ mon}^{-1}$  (Figure 2). The modeled wind fields, which are the same in all scenarios, transport the dust from source regions, leading to the enhanced concentrations across much of the domain, as seen in Figure 4."

Line 280: Again, it would be good if the authors can be specific about both vegetation change and land use change. For example, what is the changing land use in west Texas in this scenario?

We have added Figure S8 to show changes in land use under future climate. We further clarify. Lines 370-371: "We find that dust concentrations decrease only in a limited area in western Texas due to decreased pasture (Figures 3 and S9)."

Line 298: I agree with this statement about the importance of robust representation of both future vegetation changes and the sensitivity of dust emissions to these changes. However, I question whether this need has actually been addressed in the present study. See my major concerns above relating to: 1) description of changes lacking detail and grounding in actual vegetation and land use changes occurring across the southwest, and 2) physical representation of vegetation in the dust model ignores the major effect of vegetation on dust emission (lateral process) and the interactions with vegetation changes that are likely to occur.

As described above, we have attempted to address these issues in our revision. We repeat some of the revised text below.

**1. Grounding our study in actual vegetation and land use changes.**

Lines 49-55: "Southwestern North America is covered by desert grassland, perennial grassland, savanna, desert scrub, and grassy shrublands or woodlands (McClaran and Van Devender, 1997). In recent decades, a gradual transition from grasslands to shrubland has been observed across much of this region, with increased aridity, atmospheric CO2 enrichment, and livestock grazing all possibly playing a role in this trend (Bestelmeyer et al., 2018). Future climate change may further prolong this transition, especially since shrubs fare better than grasses under a climate regime characterized by large fluctuations in annual precipitation (Bestelmeyer et al., 2018; Edwards et al., 2019)."

Lines 311-314: "As predicted by previous studies (Bestelmeyer et al., 2018; Edwards et al., 2019), C3 perennial grasses (C3gr) in this case decrease across a large swath extending from Arizona through Mexico, showing the impacts of warmer temperatures and reduced precipitation, as well as (for Mexico) land use change."

Lines 414-418: "In summary, we find that as atmospheric  $CO_2$  levels rise vegetation growth is enhanced and dust mobilization decreases, offsetting the impacts of warmer temperatures and reduced rainfall, at least in some areas. These results are consistent with evidence that  $CO_2$ fertilization is already occurring in arid or semiarid environments like southwestern North America (Donohue et al., 2013; Haverd et al., 2020)."

**2. Representation of all the effects of vegetation on dust emissions.**

Lines 431-438: "The different vegetation types in our model are quantified as fractions of gridcells, which have relatively large spatial dimensions of  $\sim$ 50 km  $\times$  60 km. This means the model cannot capture the spatial heterogeneity of land cover, and the aerodynamic sheltering effects of vegetation on wind erosion are neglected, as they are in most 3-D global model studies. Such sheltering could play a large role in dust mobilization (e.g., Liu et al., 1990). New methods involving satellite observations of surface albedo promise to improve understanding of the effects of aerodynamic sheltering on dust mobilization, at least for the present-day (Chappell and Webb, 2016; Webb and Pierre, 2018)."

Lines 440-443: "In addition, as recommended by Zender et al. (2003), we apply a globally uniform surface roughness  $Z_0$  in the model, which means that the impact of changing vegetation conditions on friction velocity is not taken into account. Future work could address this weakness by varying friction velocity according to vegetation type."

Lines 448-453: "Within these limitations, our study quantifies the potential impacts of changing land cover and land use practices on dust mobilization and fine dust concentration over the coming century in southwestern North America. Our work builds on previous studies focused on future dust in this region by (1) more accurately capturing the transport of dust from source regions with a dynamical 3-D model, (2) considering results with and without CO2 fertilization, and (3) including the impact of land use trends."

In sum, although we have not been able to "close the book" on future dust emissions over the southwestern North America, our work provides an increment of progress and highlights a new threat to human health in the face of climate change.

Line 312: I think the emphasis on CO2 perhaps oversimplifies the controls. These dryland systems are largely water, not nutrient, limited. But not only cover - this will also be C3 vs C4 dominance and so the proportions and kinds of vegetation on these landscapes will influence responses to elevated CO2. Vegetation state changes today and into the future (influenced to some degree by CO2) are likely to have a far greater effect on the structure and cover of protective roughness.

As described above, we now more strongly acknowledge the limitations of this study, in particular the neglect of the variation of surface roughness lengths for different vegetation types. We also comment on the effect of climate change on  $C_3$  grasses in the model.

Lines 311-314: "As predicted by previous studies (Bestelmeyer et al., 2018; Edwards et al., 2019), C3 perennial grasses (C3gr) in this case decrease across a large swath extending from Arizona through Mexico, showing the impacts of warmer temperatures and reduced precipitation, as well as (for Mexico) land use change."

| 1  | Response of dust emissions in southwestern North America to 21 st                            |                                                            |
|----|---------------------------------------------------------------------------------------------------------|------------------------------------------------------------|
| 2  | century trends in climate, CO2 fertilization, and land use:                                             |                                                            |
| 3  | Implications for air quality                                                                            |                                                            |
| 4  | Yang Li 1 , Loretta J. Mickley 1 , Jed O. Kaplan 2                     |                                                            |
| 5  | 1 John A. Paulson School of Engineering and Applied Sciences, Harvard University, Cambridge, |                                                            |
| 6  | MA, USA                                                                                                 |                                                            |
| 7  | 2 Department of Earth Sciences, The University of Hong Kong, Hong Kong, China                |                                                            |
| 8  | Correspondence to: Yang Li (yangli@seas.harvard.edu)                                                    |                                                            |
| 9  |                                                                                                         |                                                            |
| 10 | Abstract. Climate models predict a shift toward warmer and drier environments in southwestern           |                                                            |
| 11 | North America over the 21st century. The consequences of climate change for dust mobilization           | Deleted: However, the change are sometimes of       |
| 12 | and concentrations are unknown, but could have large implications for human health, given               |                                                            |
| 13 | connections between dust inhalation and disease. Here we drive, a dynamic vegetation model (LPJ-        | Deleted: link                                              |
| 14 | LMfire) with future scenarios of climate and land use, and link the results to a chemical transport     |                                                            |
| 15 | model (GEOS-Chem) to assess the impacts of land cover on dust mobilization and fine dust                | Deleted: future change
fertilization, and land u |
| 16 | concentrations (defined as dust particles less than 2.5 microns in diameter) on surface air quality.    | Deleted: vegetation in the impacts of changing             |
| 17 | In the most extreme warming scenario (RCP8.5), we find that surface temperatures in southwestern        | Deleted: assess the net                                    |
| 18 | North America during the season of greatest dust emissions (March, April, and May) warm by 3.3          | Deleted: , and to invest                                   |
| 19 | K and precipitation decreases by nearly 40% by 2100, These conditions lead to vegetation dieback        | Deleted: in the most expring (March, April, a              |
| 20 | and an increase in dust-producing bare ground. Enhanced CO2 fertilization, however, offsets the         | emissions                                                  |
| 21 | modeled effects of warming temperatures and rainfall deficit on vegetation in some areas of the         |                                                            |
| 22 | southwestern United States. Considering all three factors in RCP8.5 scenario, dust concentrations       |                                                            |
|    |                                                                                                         |                                                            |

projected dust trends under climate contradictory

es in three factors – climate, CO2 use practices – this region. From there we investigate g vegetation on

effect on

tigate the consequences for

extreme warming scenario (RCP8.5) in and May), the season of greatest dust

[revised manuscript text omitted]

dust burden (e.g., Harrison et al., 2001, Mahowald and Luo, 2003 and Mahowald et al., 2006).

These estimates of future dust emissions depended in large part on the choice of model applied, as

between observed present-day dust concentrations and meteorological conditions or leaf area index

(LAI). Hand et al., 2016 found that fine dust concentrations in spring in this region correlated with

In southwestern North America, a few recent studies examined statistical relationships

127

128 129

130

131

132

demonstrated by Tegen et al., 2004.

**Moved down [1]:** To investigate the potential effects of climate change, increasing CO2 concentrations, and future land use practices on dust mobilization in southwestern North America, we couple a dynamic vegetation model (LPJ-LMfire) to a chemical transport model (GEOS-Chem) and perform a series of experiments in scenarios of future environmental conditions.

4

| 140 | the Pacific Decadal Oscillation (PDO), indicating the importance of large-scale climate patterns in  |                                   |
|-----|------------------------------------------------------------------------------------------------------|-----------------------------------|
| 141 | the mobilization and transport of regional fine dust. Tong et al., 2017 further determined that the  |                                   |
| 142 | observed 240% increase in the frequency of windblown dust storms from 1990s to 2000s in the          |                                   |
| 143 | southwestern United States was likely associated with the PDO. Similarly, Achakulwisut et al.,       |                                   |
| 144 | 2017 found that the 2002-2015 increase in average March fine dust concentrations in this region      |                                   |
| 145 | was driven by a combination of positive PDO conditions and phase of the El Nino-Southern             |                                   |
| 146 | Oscillation. More recently, Achakulwisut et al., 2018 identified the Standardized Precipitation-     |                                   |
| 147 | Evapotranspiration Index as a useful indicator of present-day dust variability. Applying that metric |                                   |
| 148 | to an ensemble of future climate projections, these authors predicted increases of 26-46% in fine    |                                   |
| 149 | dust concentrations over the U.S. Southwest in spring by 2100. In contrast, Pu and Ginoux, 2017      |                                   |
| 150 | found that the frequency of extreme dust days decreases slightly in spring in this region due to     |                                   |
| 151 | reduced extent of bare ground under 21st century climate change.                                     | Deleted                           |
| 152 | These regional studies relied mainly on statistical models that relate local and/or large scale      | Deleted                           |
| 153 | meteorological conditions to dust emissions in southwestern North America. Pu and Ginoux, 2017       |                                   |
| 154 | also considered changing LAI in their model, but these dust-LAI relationships were derived from      |                                   |
| 155 | a relatively sparse dataset, casting some uncertainty on the results (Achakulwisut et al., 2018). In |                                   |
| 156 | this study, we investigate the effects of climate change, increasing CO2 fertilization, and future   | Deleted                           |
| 157 | land use practices on vegetation in southwestern North America, and we examine the response of       | cilliate                          |
| 158 | dust mobilization due to these changes in vegetation. With regard to climate, we examine whether     |                                   |
| 159 | a shift to warmer, drier conditions by 2100 enhances dust mobilization in this region by reducing    |                                   |
| 160 | vegetation cover and exposing bare land, To that end, we couple the LPJ-LMfire dynamic               | Moved                             |
| 161 | vegetation model to the chemical transport model GEOS-Chem to study vegetation dynamics and          | Deleted
increasir
on dust i |
| 162 | dust mobilization under different conditions and climate scenarios, allowing consideration of        | Deleted                           |

: land

: previous

Le examine the response of dust mobilization due to -induced changes in vegetation,

: across the 21st century

**(insertion) [1]**

I: investigate the potential effects of climate change, ing CO2 concentrations, and future land use practices mobilization in southwestern North America

| Deleted. | many |
|----------|------|
| Deleteu. | man  |

| 175 | several factors driving future dust mobilization in the southwestern North America. We focus on     |
|-----|-----------------------------------------------------------------------------------------------------|
| 176 | fine dust particles in springtime (March, April, and May), because it is the season of highest dust |
| 177 | concentrations in the southwestern U.S. (Hand et al., 2017). Given the deleterious impacts of       |
| 178 | airborne dust on human health, our dust projections under different climate scenarios have value    |
| 179 | for understanding the full array of potential consequences of anthropogenic climate change.         |
| 180 |                                                                                                     |

**181 2 Methods**

| 182 | We examine dust mobilization in southwestern North America, here defined as $25^{\circ}N$ –                                        |
|-----|------------------------------------------------------------------------------------------------------------------------------------|
| 183 | $37^{\circ}N$ , $100^{\circ}W - 115^{\circ}W$ (Figure 1), during the late- $21^{st}$ century under scenarios of future climate     |
| 184 | and land use based on two Representative Concentration Pathways (RCPs). RCP4.5 and RCP8.5                                          |
| 185 | capture two possible climate trajectories over the 21st century, beginning in 2006. RCP4.5                                         |
| 186 | represents a scenario of moderate future climate change with gradual reduction in greenhouse gas                                   |
| 187 | (GHG) emissions after 2050 and a radiative forcing at 2100 relative to pre-industrial values of +4.5                               |
| 188 | W m -2 , while RCP8.5 represents a more extreme scenario with continued increases in GHGs                               |
| 189 | throughout the $21^{st}$ century and a radiative forcing of +8.5 W m -2 at 2100. For each RCP, we                       |
| 190 | investigate the changes in vegetation for three cases: 1) an all-factor case that includes changes in                              |
| 191 | climate, land use, and CO 2 fertilization; 2) a fixed-CO 2 case that includes changes in only climate |
| 192 | and land use; and 3) a fixed-land use case that includes changes in only climate and $\mathrm{CO}_2$                               |
| 193 | fertilization.                                                                                                                     |

We use LPJ-LMfire, a dynamic global vegetation model, to estimate changes in vegetation
 under future conditions (Pfeiffer et al., 2013). Meteorology to drive LPJ-LMfire is taken from the
 Goddard Institute for Space Studies (GISS) climate model (Nazarenko et al., 2015). Using the
 GEOS-Chem emission component (HEMCO), we then calculate dust emissions based on the LPJ-

| Deleted: following | ) |
|--------------------|---|
| Deleted: scenarios |   |
| Deleted: scenario  |   |
| Deleted: scenario  |   |
| Deleted: scenario  |   |

**Deleted:** RCP4.5 represents a moderate pathway with gradual reduction in greenhouse gas (GHG) emissions after 2050, while RCP8.5 assumes continued increases in GHGs throughout the 21st century. To estimate changes in vegetation under future meteorological conditions, w

**Moved down [2]:** Present-day and future meteorological fields, including surface temperature and precipitation, are simulated by the Goddard Institute for Space Studies (GISS) Model E climate model, and these are fed into LPJ-LMfre. For each RCP, we investigate the changes in vegetation following three scenarios: 1) an all-factor scenario that includes changes in climate, land use, and CO2 fertilization; 2) a fixed-CO2 scenario that includes changes in only climate and land use; and 3) a fixed-land use scenario that includes changes in only climate and CO2 fertilization.

| 221 | generated vegetation area index (VAI) for all scenarios. We apply the resulting dust emissions to                           |                                          | Deletee                                   |
|-----|-----------------------------------------------------------------------------------------------------------------------------|------------------------------------------|-------------------------------------------|
| 222 | the global chemical transport model GEOS. Chem to simulate the distribution of fine dust across                             |                                          | Delete                                    |
| 222 | the global element transport model OLOS-chem to simulate the distribution of the dust deloss                                |                                          | Deletee                                   |
| 223 | the southwestern North America.                                                                                             |                                          | Deletee                                   |
| 224 |                                                                                                                             |                                          | Deleted
longitu
the peri
For cor |
| 225 | 2.1 GISS Model E                                                                                                            |                                          | simulat
21 st cer           |
| 226 | Present-day and future meteorological fields for RCP4.5 and RCP8.5 are simulated by the                                     |                                          | years (2
meteor
allows              |
| 227 | GISS Model E climate model (Nazarenko et al., 2015), configured for Phase 5 of the Coupled                                  |                                          | dust mo
section
simulat             |
| 228 | Model Intercomparison Project (CMIP5; https://esgf-node.llnl.gov/search/cmip5/, last accessed on                            |                                          | Moved
slight in                        |
| 229 | 17 July 2020). The simulations cover the years 1801 to 2100 at a spatial resolution of 2° latitude x                        | annan an a | tempera
over the
(2095-2            |
| 230 | 2.5° longitude. Changes in climate in the GISS model are driven by increasing greenhouse gases.                             |                                          | RCP8.5                                    |
| 231 | In RCP4.5, CO 2 concentrations increase to 550 ppm by 2100; in RCP8.5 the CO 2 increases to 1960      |                                          | in temp
present                        |
| 232 | ppm ((Meinshausen et al., 2011),                                                                                            |                                          | Moved                                     |
| 233 | Junder RCP4.5, the GISS model predicts a slight increase of 0.45 K in springtime mean                                       |                                          | Delete                                    |
| 224 | surface temperatures and an increase in mean precipitation by $-17\%$ over the southwestern North                           |                                          | Delete                                    |
| 234 | surface temperatures and an increase in mean precipitation by $\sim 1770$ over the southwestern North                       |                                          | Deletee                                   |
| 235 | America by the 2100 time slice (2095-2099), relative to the present day (2011-2015). In contrast,                           |                                          | Deleted
vegetat                        |
| 236 | under RCP8.5, the 5-year mean springtime temperature increases significantly by 3.29 K by 2100                              |                                          | fertiliza
only cli                     |
| 237 | and mean precipitation decreases by ~39%. The spatial distributions of the changes in temperature                           |                                          | Moved                                     |
| 238 | and precipitation by 2100 under RCP8.5 are presented in the Supplement (Figure S2). In addition,                            |                                          | Deleter                                   |
| 239 | lightning strike densities decrease by ~0.006 strikes km -2 d -1 over Arizona in RCP4.5, but increase |                                          | Deleter                                   |
| 240 | by the same magnitude in this region in RCP8.5 (Li et al., 2020). Lightning strikes play a major                            |                                          |                                           |
| 241 | role for wildfire ignition in this region, while wildfires may influence landscape succession (e.g.,                        |                                          |                                           |
| 242 | Bodner and Robles, 2017). Finally, future surface wind speeds do not change significantly under                             |                                          |                                           |
| 243 | RCP4.5, but increase slightly by ~4% across southwestern North America under RCP8.5 by 2100                                 |                                          |                                           |

**d: in different d: then**

**d: these**

**d: and**

d: at a spatial resolution of 0.5° latitude x 0.625° de. For each RCP, the LPJ-LMfire simulation covers iod 2006-2100 continuously, with monthly resolution. nputational reasons, we limit our GEOS-Chem tions to two time-slices centered on the early and late ntury, with each time slice covering 5 continuous 2011-2015 and 2095-2099). We apply present-day ology to both time slices in GEOS-Chem, which us to focus on the effect of changing land cover on obilization. More information is in the Methods , including validation of the GEOS-Chem dust tion for the present-day.

down [3]: Under RCP4.5, the GISS model predicts a ncrease of 0.45 K in springtime mean surface atures and an increase in mean precipitation by ~17% e southwestern North America by the 2100 time slice 2099), relative to the present day (2011-2015). Under 5, the 5-year mean springtime temperature increases cantly by 3.29 K by 2100 and mean precipitation ses by  $\sim$ 39%. The spatial distributions of the changes perature and precipitation by 2100 under RCP8.5 are ed in the Supplement (Figure S1).

**(insertion) [2]**

d: , including surface temperature and precipitation, d: Goddard Institute for Space Studies (

**d: )**

**d:, and these are fed into LPJ-LMfire**

**d:** For each RCP, we investigate the changes in tion following three scenarios: 1) an all-factor scenario ludes changes in climate, land use, and CO2 tion; 2) a fixed-CO2 scenario that includes changes in mate 
[revised manuscript text omitted]

413

403

407

$$A_V = \min \left[ 1.0, \min(VAI, VAI_t) / VAI_t \right]$$

**414 where $VAI_t$ is the threshold for complete suppression of dust emissions, set here to 0.3 m2 m-2 415 (Zender et al., 2003; Mahowald et al., 1999).**

416 To compute the dust fluxes, we need to convert LAI from LPJ-LMfire to VAI. VAI is 417 generally defined as the sum of LAI plus stem area index (SAI). Assuming immediate removal of 418 all dead leaves, the fractional vegetation cover,  $\sigma_{v}$  can be used to represent SAL for the different 419 PFTs (Zeng et al., 2002). Given that the threshold  $VAI_t$  for no dust emission is relatively low (0.3 420  $m^2 m^{-2}$ ), leaf area dominates stem area in the suppression of dust mobilization in the model. In 421 areas where LAI is greater than SAI, we therefore assume that SAI does not play a role in 422 controlling dust emissions, and we set LAI equivalent to VAI. We also assume that C3 and C4 423 grasses have zero stem area to avoid overestimating VAI during the winter and early spring when 424 such grasses are dead. Based on the method of Zeng et al., 2002, with modifications, we calculate 425 VAI in each grid cell as

**Deleted: It**

| 1                         | Deleted: as                                                                                                                                                                                                                                                                                                                                                                                                                          |
|---------------------------|--------------------------------------------------------------------------------------------------------------------------------------------------------------------------------------------------------------------------------------------------------------------------------------------------------------------------------------------------------------------------------------------------------------------------------------|
|                           | Deleted: and is                                                                                                                                                                                                                                                                                                                                                                                                                      |
|                           | Deleted: as                                                                                                                                                                                                                                                                                                                                                                                                                          |
| λ                         | Moved (insertion) [5]                                                                                                                                                                                                                                                                                                                                                                                                                |
|                           | Moved up [4]: LPJ-LMfire calculates the monthly leaf area indices (LAI) and fractional vegetation cover () for nine plant functional types (PFTs): tropical broadleaf evergreen, tropical broadleaf raingreen, temperate needleleaf evergreen, temperate broadleaf avergreen, temperate broadleaf evergreen, boreal needleleaf evergreen, and boreal summergreen trees, as well as C 3 and C 4 grasses. |
| $\left( \right)$          | Deleted: the                                                                                                                                                                                                                                                                                                                                                                                                                         |
| $\langle \rangle \rangle$ | Deleted: as in Sellers et al., 1996,                                                                                                                                                                                                                                                                                                                                                                                                 |
| $\langle \rangle \rangle$ | Deleted: stem area index (                                                                                                                                                                                                                                                                                                                                                                                                           |
| (())                      | Deleted: )                                                                                                                                                                                                                                                                                                                                                                                                                           |
|                           | Moved up [5]: VAI is generally defined as the sum of the LAI plus SAI.                                                                                                                                                                                                                                                                                                                                                               |
|                           | Deleted: VAI is generally defined as the sum of the LAI plus SAI. As VAI is generally defined as the sum of the LAI plus SAI.                                                                                                                                                                                                                                                                                                 |
|                           | Deleted: i.e.,                                                                                                                                                                                                                                                                                                                                                                                                                       |
| (  // .                   | Deleted: has a dominant                                                                                                                                                                                                                                                                                                                                                                                                              |
| ////                      | Deleted: effect on                                                                                                                                                                                                                                                                                                                                                                                                                   |
|                           | Deleted: that                                                                                                                                                                                                                                                                                                                                                                                                                        |
|                           | Deleted: is                                                                                                                                                                                                                                                                                                                                                                                                                          |
| Ì                         | Deleted: non-growing season                                                                                                                                                                                                                                                                                                                                                                                                          |
|                           |                                                                                                                                                                                                                                                                                                                                                                                                                                      |

(2),

[revised manuscript text omitted]